



# Controls on the relative abundances and rates of nitrifying microorganisms in the ocean

Emily J. Zakem[1,2], Barbara Bayer[3], Wei Qin[4], Alyson Santoro[5], Yao Zhang[6], and Naomi M. Levine[2]

[1]Department of Global Ecology, Carnegie Institution for Science, Stanford, CA USA
[2]Department of Biological Sciences, University of Southern California, Los Angeles, CA USA
[3]Department of Microbiology and Ecosystem Science, University of Vienna, Vienna, Austria
[4]Department of Microbiology and Plant Biology, University of Oklahoma, Norman, OK USA
[5]Department of Ecology, Evolution and Marine Biology, University of California, Santa Barbara, CA USA
[6]State Key Laboratory of Marine Environmental Science and College of Ocean and Earth Sciences, Xiamen University, Xiamen, China

**Correspondence:** Emily J. Zakem (ezakem@carnegiescience.edu)

**Abstract.** Nitrification controls the oxidation state of bioavailable nitrogen. Distinct clades of chemoautotrophic microorganisms – predominantly, ammonia-oxidizing archaea (AOA) and nitrite-oxidizing bacteria (NOB) – regulate the two steps of nitrification in the ocean, but explanations for their observed relative abundances and nitrification rates remain incomplete, and their contributions to the global marine carbon cycle via carbon fixation remain unresolved. Using a mechanistic microbial

ecosystem model with nitrifying functional types, we derive simple expressions for the controls on AOA and NOB in the deep, oxygenated open ocean. The relative yields, loss rates, and cell quotas of AOA and NOB control their relative abundances, though we do not need to invoke a difference in loss rates to explain the observed relative abundances. The supply of ammonium, not the traits of AOA or NOB, controls the relatively equal ammonia- and nitrite-oxidation rates at steady state. The relative yields of AOA and NOB alone set their relative bulk carbon fixation rates in the water column. The quantitative rela-

tionships are consistent with multiple *in situ* datasets. In a complex global ecosystem model, nitrification emerges dynamically across diverse ocean environments, and ammonia and nitrite oxidation and their associated carbon fixation rates are decoupled due to physical transport and complex ecological interactions in some environments. Nevertheless, the simple expressions capture global patterns to first order. The model provides a mechanistically estimated upper bound on global chemoautotrophic carbon fixation of 0.2–0.5 Pg C yr$^{-1}$, which is on the low end of the wide range of previous estimates. Modeled carbon fixation

by NOB (about 0.1 Pg C yr$^{-1}$) is substantially lower than by AOA (0.2–0.3 Pg C yr$^{-1}$), predominantly reflecting the relative yields. The simple expressions derived here can be used to quantify the biogeochemical impacts of additional metabolic pathways (i.e. mixotrophy) of nitrifying clades and to identify alternative carbon-fixing metabolisms in the deep ocean.

## 1 Introduction

Remineralizing organisms control nutrient cycling and carbon storage in the biosphere. Organic nitrogen is remineralized and

oxidized in sequential steps, each carried out by distinct groups of organisms. Heterotrophs oxidize organic carbon for energy and typically excrete nitrogen in simplified, reduced forms, such as urea and ammonium ($NH_4^+$, here referred to interchangeably





with ammonia, $NH_3$). Excretion of reduced nitrogen in shallow, sunlit waters predominantly resupplies primary production locally. At depth, remineralization maintains the marine "biological pump" of carbon (Volk and Hoffert, 1985), and the excreted nitrogen is oxidized by chemoautotrophic nitrifying microorganisms to nitrate ($NO_3^-$), which fills the deep ocean.

In the ocean and most aquatic environments, nitrification is a two-step process carried out by two distinct microbial clades: ammonia-oxidizing archaea (AOA) and nitrite-oxidizing bacteria (NOB) (Ward et al., 2008). AOA are the most ubiquitous chemoautotrophs in the dark ocean (Karner et al., 2001; Wuchter et al., 2006; Santoro et al., 2019), due in part to their small cell size (Könneke et al., 2005; Santoro and Casciotti, 2011). Larger NOB are equally widespread but less numerous (Santoro et al., 2010; Pachiadaki et al., 2017; Santoro et al., 2019), though the NOB metalloenzyme nitrite oxidoreductase has been shown to
be one of the most abundant proteins in the mesopelagic ocean (Saito et al., 2020). Both chemoautotrophic metabolisms are much less efficient than photoautotrophy, though their underlying redox reactions suggest that ammonia oxidation should yield more biomass than nitrite oxidation.

Our understanding of the global-scale biogeochemical roles of AOA and NOB remains incomplete. The role of NOB in the global carbon cycle in particular remains unclear. Currently, estimates of global carbon fixation by NOB range over an order
of magnitude (Pachiadaki et al., 2017; Zhang et al., 2020; Bayer et al., 2022). Recent studies demonstrate that some types of NOB are metabolically diverse, with the abilities to break down urea, oxidize compounds other than $NO_2^-$, and reduce $NO_3^-$ in addition to $O_2$ (Koch et al., 2014, 2015; Füssel et al., 2017; Bayer et al., 2020), though the large-scale biogeochemical impacts of this versatility are also unclear.

In order to anticipate present and future biogeochemical impacts of nitrifying microorganisms, we must better understand
the controls on their abundances and rates. Observations show AOA at consistently higher abundances (7- to 11-fold) than NOB in the water column (Fig. 2a; Santoro et al. (2010, 2019); Zhang et al. (2020)), yet relatively equal rates of $NH_3$ and $NO_2^-$ oxidation (Dore and Karl (1996); Ward et al. (2008); Zhang et al. (2020)). What controls these patterns? Second, how do these nitrification rates relate to *in situ* chemoautotrophic carbon fixation rates across scales, from cellular to global levels? Here, we use a mechanistic ecosystem model to interpret observations of the nitrification system in new ways, quantitatively explain the
relationships between nitrifier abundances and their transformations of N and C in the water column, and connect micro-scale observations mechanistically to global-scale estimates.

Previous work constructed a theoretically grounded ecological model of nitrifying populations that is useful for large-scale biogeochemical modeling (Zakem et al., 2018). The model resolves remineralization explicitly with dynamic populations of heterotrophic and nitrifying microorganisms (Fig. 1). Redox-informed, cellular-level metabolic budgets relate nutrient uptake
and excretions of waste products to biomass synthesis rates. When incorporated into an ocean biogeochemical model, the locations of nitrification emerge dynamically from ecological interactions among the microbial populations. Competition between nitrifiers and phytoplankton for $NH_4^+$ and $NO_2^-$ results in the common, but not exclusive, restriction of nitrification from the sunlit surface.

Here, using this ecosystem model, we first provide simple, mechanistic expressions for the relative abundances, nitrification
rates, and carbon fixation rates of AOA and NOB in the dark, oxygenated ocean. This allows us to explain patterns exhibited in multiple sets of *in situ* observations using just a few parameters. Second, we examine how these expressions become relevant





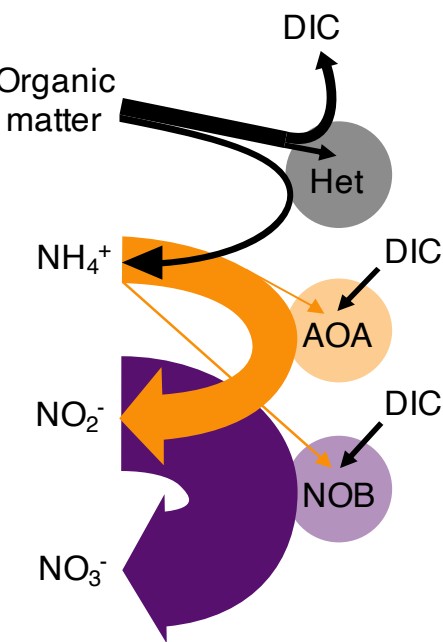

**Figure 1.** Schematic of the nitrogen remineralization sequence driven by microbial functional types. Organic matter is consumed by heterotrophic organisms (Het), with ammonium ($NH_4^+$) and dissolved inorganic carbon (DIC) as waste products. Chemoautotrophic ammonia-oxidizing archaea (AOA) consume $NH_4^+$ and excrete nitrite ($NO_2^-$). Chemoautotrophic nitrite-oxidizing bacteria (NOB) consume $NO_2^-$ and excrete nitrate ($NO_3^-$). AOA and NOB may assimilate $NH_4^+$ (as illustrated) or simple organic nitrogen compounds. Straight arrows indicate the substrates used for biomass synthesis, while horseshoe-shaped arrows indicate respiration substrates and products. The widths of each of the arrows corresponds to the metabolic budgets used in this study. For example, NOB require significantly more $NO_2^-$ than the amount of $NH_4^+$ required by AOA to produce the same amount of biomass. Though not indicated, heterotrophs may also fix DIC in the dark ocean.

with depth by comparing a high-resolution, dynamic water column model to observations. Third, we investigate global scale relationships using a three-dimensional, global configuration of the ecosystem model.

## 2 The model

We employ a marine ecosystem model that resolves the growth, respiration, and mortality of ammonia- and nitrite-oxidizing biomass, representing aggregated populations of AOA and NOB (Zakem et al., 2018). The model also resolves phytoplankton, heterotrophic bacteria, and zooplankton biomasses and inorganic nitrogen concentrations. Respiration by the microbial populations (including zooplankton) constitutes all of the remineralization of organic matter back into its inorganic constituents. Temperature modifies the metabolic rates of all populations, as detailed in Appendix A. In the Appendix, we provide the full

set of model equations as well as detailed descriptions of the configurations and parameter values. Here, we present: 1) the key model equations describing nitrification, 2) an overview of the different model configurations, and 3) the key parameter values used to describe and differentiate the two nitrifying populations and the treatment of their uncertainty.





## 2.1 Key equations

Following Zakem et al. (2018), the growth rates of the ammonia-oxidizing archaeal ($\mu_{AOA}$) and nitrite-oxidizing bacterial

($\mu_{NOB}$) functional types are calculated from their biomass yields and uptake rates of reduced nitrogen (assumed to be the limiting nutrients) as:

$$\mu_{AOA} = y_{\mathrm{NH_4}} V_{maxN} \frac{\mathrm{NH_4^+}}{\mathrm{NH_4^+} + K_N} \tag{1}$$

$$\mu_{NOB} = y_{\mathrm{NO_2}} V_{maxN} \frac{\mathrm{NO_2^-}}{\mathrm{NO_2^-} + K_N}, \tag{2}$$

where $y_{\mathrm{NH_4}}$ and $y_{\mathrm{NO_2}}$ are the biomass yields of AOA and NOB, respectively (as mol biomass synthesized per mol $\mathrm{NH_4^+}$ or

$\mathrm{NO_2^-}$ utilized), $V_{maxN}$ (mol $\mathrm{NH_4^+}$ or $\mathrm{NO_2^-}$ per mol biomass) is the specific maximum uptake rate, and $K_N$ (mol $\mathrm{NH_4^+}$ or $\mathrm{NO_2^-}$ $\mathrm{L^{-1}}$) is the half-saturation concentration for uptake. Here, we assume that AOA and NOB populations have similar uptake kinetics in the deep ocean, following the results of Zhang et al. (2020), though we demonstrate below that the uptake kinetics do not have a significant impact on the results of this study. Each state variable in the model (tracer $C$) is transported by the ocean circulation according to velocities $\boldsymbol{u}$ and diffusion coefficients $\boldsymbol{\kappa}$ as:

$$\frac{\partial C}{\partial t} = -\nabla \cdot (\mathbf{u}C) + \nabla \cdot (\boldsymbol{\kappa} \nabla C) + S_C \tag{3}$$

where $S_C$ are additional sources and sinks. For biomass concentrations ($B_{AOA}$ and $B_{NOB}$; mol N $\mathrm{L^{-1}}$) and DIN concentrations (mol N $\mathrm{L^{-1}}$):

$$S_{B_{AOA}} = B_{AOA}(\mu_{AOA} - L_{AOA}) \tag{4}$$

$$S_{B_{NOB}} = B_{NOB}(\mu_{NOB} - L_{NOB}) \tag{5}$$

$$S_{\mathrm{NH_4^+}} = -\frac{1}{y_{\mathrm{NH_4}}}\mu_{AOA}B_{AOA} - \mu_{NOB}B_{NOB} - V_{\mathrm{NH_4}}P + e_{NH_4}B_{het} + e_{NH_4}Z \tag{6}$$

$$S_{\mathrm{NO_2^-}} = (\frac{1}{y_{\mathrm{NH_4}}-1})\mu_{AOA}B_{AOA} - \frac{1}{y_{\mathrm{NO_2}}}\mu_{NOB}B_{NOB} - V_{\mathrm{NO_2}}P \tag{7}$$

$$S_{\mathrm{NO_3^-}} = \frac{1}{y_{\mathrm{NO_2}}}\mu_{NOB}B_{NOB} - V_{\mathrm{NO_3}}P \tag{8}$$

where $L$ ($\mathrm{t^{-1}}$) is the specific biomass loss rate function and $V_{\mathrm{NH_4}}$, and $V_{\mathrm{NO_2}}$, $V_{\mathrm{NO_3}}$ are uptake of DIN by phytoplankton $P$. Excretion of $\mathrm{NH_4^+}$ ($e_{\mathrm{NH_4}}$) by heterotrophic bacteria ($B_{het}$) and zooplankton ($Z$) supplies $\mathrm{NH_4^+}$ according to their growth

efficiencies. The biomass losses of all populations supply the dissolved and particulate pools of organic matter that are remineralized by the heterotrophs. Loss rate $L$ represents biomass losses to grazing, viral lysis, maintenance, and senescence, and is a function of both the population's biomass and that of zooplankton predator $Z$ (Appendix A). In the model versions presented, we assume one microzooplankton grazer preys on all non-photoautotrophic microbial populations ($B_{het}$, $B_{AOA}$, and $B_{NOB}$).



The actual food web configuration is not known, and different configurations are possible, and so our uncertainty estimates
account for a wide range in variation of loss rates between the populations. We assume that both AOA and NOB consume
$NH_4^+$ (or simple organic compounds such as urea) for assimilation into biomass synthesis. Because these assimilation terms
are small relative to the other terms due to the low nitrifier biomass yields (i.e. $y^{-1} >> 1$), they are negligible in the solutions.
Light inhibition is not imposed upon the nitrifying populations. Rather, the restriction of nitrification from the sunlit surface
emerges as a consequence of ecological interactions (Zakem et al., 2018).

## 2.2   Model configurations

We use a hierarchy of configurations of the ecosystem model to answer our research questions. First, we use a set of equations
that are simplified to represent the dynamics in the dark, oxygenated ocean, neglecting the impacts of phytoplankton and phys-
ical transport. We examine the steady state balances of at a single point. This allows us to develop simple, linear expressions
for the relative abundances and rates of AOA and NOB functional types as functions of just a few parameters. Appendix C
provides the simplified set of equations and the derivation of these expressions.

Second, we use a vertical water column model of the full, dynamic ecosystem (with phytoplankton, heterotrophic bacteria,
zooplankton, and physical transport) to compare the results to observations from the Western Pacific Ocean. This allows us
to evaluate and visualize how the simple expressions from the point balances become relevant at depth. Attenuation of light
and mixing with depth provide the physical structure of the 2000m stratified water column (Appendix B). We assume that
oxygen and micronutrients are abundant so that $NH_4^+$ and $NO_2^-$ limit the growth of AOA and NOB, respectively. Nitrogen is
conserved over the domain. Equations are integrated forward in time until an equilibrium state is reached. Because the model
resolves nitrogen-based biomass, we convert the biomass yields and elemental quotas using elemental ratio $R_{NC}$ from the
measured C:N contents of AOA and NOB (Bayer et al. (2022); Table A1). To quantify model uncertainty, we randomly draw
parameter values from ranges in yields, loss rate parameters, and cell quotas of the AOA and NOB functional types to construct
an ensemble of 2000 equilibrium model solutions. We illustrate the range between the 5th and 95th percentiles of the ensemble.
Table A1 lists all parameter values, including the ranges used for the ensemble. Unless noted as Gaussian, uniform distributions
are used.

Third, we analyze global-scale relationships using a 3D global configuration of the ecosystem model. The global model
allows us to examine whether the simple point balances are relevant across the diverse environments of the global ocean and
to estimate globally integrated rates. The nitrification ecosystem model is integrated with the Darwin-MITgcm model and
coupled to the ECCO-GODAE state estimate of the ocean circulation ($1° × 1°$ horizontal resolution; 23 vertical levels; Follows
et al. (2007); Dutkiewicz et al. (2015b); Wunsch and Heimbach (2007)). The ecosystem component resolves the cycling
of carbon, nitrogen, phosphorus, iron, and silica. AOA and NOB growth are limited by oxygen, according to redox-based
respiration budgets, as well as phosphorus and iron requirements for biomass synthesis, as in Zakem et al. (2018). In addition
to the AOA and NOB populations, we resolve six phytoplankton populations, four zooplankton populations, two heterotrophic
bacteria types, and multiple anaerobic heterotrophic (nitrate-reducing and denitrifying) and chemoautotrophic (anammox)
metabolic functional types. The configuration is identical to that of Zakem et al. (2018) except for: (1) the incorporation of



the recently measured biomass yields from Bayer et al. (2022), which are higher than the previous model input values and so result in higher nitrifier biomass and C fixation rates, (2) the assumption of equal uptake kinetic parameters for AOA and

NOB following the results of Zhang et al. (2020), which increases the competitive ability of NOB against phytoplankton for DIN in the euphotic zone and so increases nitrite oxidation rates slightly there, and (3) the assumption that the metabolic rates of nitrifying microorganisms are sensitive to temperature using the same temperature sensitivity function as for the other microbial populations. This impacts the global estimates only slightly, well within our reported uncertainty range. (In Zakem et al. (2018), nitrifier metabolic rates were not modified by temperature following the empirical results of (Horak et al., 2013).)

These three exceptions is also included in the water column model configuration.

We estimate a range of globally integrated rates by incorporating a range of AOA and NOB yields and loss rate parameters into multiple global simulations (Table 1, Table A1). After sensitivity tests, we constructed three global model versions: one using the default parameter values, one in which the range of nitrifier parameter values gave the lowest estimate of the rates (specifically, the lower estimates of the yields and the higher estimates of loss rate parameters for both AOA and NOB), and

one in which they gave the highest estimate of the rates (specifically, the upper estimates of the yields and the lower estimates of the loss rate parameters). For the grazing rates, we varied the grazing palatability coefficient, which modulates the rate of grazing on each prey population individually. We initialized the global model with climatological nutrient concentrations and the default parameter values. Branching from a 200 year integration, the three versions are each integrated for an additional 50 years. We checked the output at intermediate timepoints to assure that the nitrification and carbon fixation rates reach an

additional quasi-steady state by the 50 years.

Additional uncertainty in global nitrification and associated carbon fixation rates exists due to uncertainty in the flux of organic matter exported out of the sunlit surface. As we later clarify in our results, absolute nitrification rates are predominantly set by this export flux via the supply of $NH_4^+$ from its remineralization. Therefore, we treat the uncertainty due to the export flux by considering that the export flux in our global model is larger than previous estimates, as detailed below in section 3.3.4

(using the heterotrophic activity parameterization of Zakem et al. (2018)). Across the three global model versions, the export flux remained the same with respect to the degree of accuracy represented in Table 3. (The range of export values reported reflects the choice of cutoff of unrealistically high values in coastal waters.) Therefore, because the model provides an upper bound on the export flux, it also provides an upper bound on nitrification rates. This is a pragmatic approach because the resulting modeled global rates are lower than many previous estimates. Thus the results provide a meaningful constraint on

these rates.

We choose to use the water column model to compare to the observed water column profiles, rather than evaluating the relevant grid points from the global model, because 1) the global model is appropriate for exploring and understanding large-scale patterns, but its solutions do not precisely match the dynamics at a particular single location, and 2) the water column model allows us to efficiently equilibrate model solutions to 2000 m depth at a higher resolution. While the 3D global model

captures patterns in the balance between net primary production and remineralization broadly, it does not accurately resolve the export flux at any one location. Because improving model resolution of the export flux is beyond the scope of this study, in the water column model, we calibrate the parameters that control the sinking flux of organic matter (specifically, the parameters





setting heterotrophic bacteria, POM, and DOM) so that the $NH_4^+$ supply rate at depth is consistent with the observed profiles. This allows the aspects of the nitrification system relevant to this study to emerge dynamically in the solutions.

## 2.3 Parameter values

We parameterize the model of the nitrifying populations using a best estimate (default) set of parameter values and their uncertainties. The relative values and ranges are summarized in Table 1 (see Table A1 for absolute values and ranges). For the biomass yields and cell quotas, we use recently published measurements for AOA and NOB grown in environmentally relevant conditions: natural seawater at 15°C and 1 μM substrate (Tables 1 and 2 in Bayer et al. (2022)). Specifically, for AOA, we incorporate the average and standard deviation of the measured C fixation yields and carbon quotas of two marine-relevant organisms: *Ca.* Nitrosopelagicus U25 and *Nitrosopumilus* sp. CCS1. For NOB, we use the measured C fixation yield and carbon content of *Nitrospina* sp. Nb-3. This provides the following parameters for the yields, considering both the carbon assimilated into biomass and any excreted in dissolved form, and quotas: for AOA, a yield of 0.098±0.021 mol C fixed per mol $NH_3$ oxidized and cell quota of 11.5±2.0 fg C per cell, and for NOB, a yield of 0.043±0.004 mol C fixed per mol $NO_2^-$ oxidized and a cell quota of 39.8±11.2 fg C per cell. The yield for *Nitrospina* is higher than many previous studies because it was enhanced by growth in natural seawater and because the study accounted for the fact that C fixation lagged behind $NO_2^-$ oxidation (Bayer et al., 2022).

Differences in the loss rates between AOA and NOB populations in the ocean are not well known. Recent studies have suggested differences in opposing directions. Zhang et al. (2020) inferred that AOA loss rates are higher than those of NOB, while Kitzinger et al. (2020) inferred that NOB loss rates are higher than those of AOA. Given this uncertainty, we assume equal default mortality parameters so that the specific loss rates ($L$) of AOA and NOB are equal in the equilibrated solutions. (Note that this results in the biomass-dependent loss rates ($LB$) of AOA exceeding those of NOB in proportion to the resulting differences in biomass.) We then consider a wide range in loss rates between AOA and NOB by incorporating a $\frac{1}{3}$-fold to 3-fold relative difference in the mortality parameters of AOA and NOB in all model configurations. This is the magnitude of the difference in loss rates inferred in Zhang et al. (2020).

## 3 Results

### 3.1 Simple expressions

We derive expressions that relate the rates and abundances of AOA and NOB functional type populations as simple, yet mechanistic, functions of a few parameters (with derivations in Appendix C). Table 2 summarizes the factors impacting the relationships. Here, we show that when incorporating our estimates of parameter values, these expressions are consistent with observations in the dark, oxygenated ocean.



**Table 1.** Relative parameter values used to describe marine AOA and NOB functional types. See Table A1 for absolute values.

| Relative parameters | | Value | Reference |
|---|---|---|---|
| **Yields** ($y_{\mathrm{NH_4}} : y_{\mathrm{NO_2}}$) | | | |
| | Default | 2.3 | Bayer et al. (2022) |
| | Max | 3.1 | Bayer et al. (2022) |
| | Min | 1.6 | Bayer et al. (2022) |
| **Loss rates** ($L_{AOA} : L_{NOB}$) | | | |
| | Default | 1 | Zakem et al. (2018) |
| | Max | 3 | Zhang et al. (2020) |
| | Min | 1/3 | (Symmetrical) |
| **Quotas** ($Q_{NOB} : Q_{AOA}$) | | | |
| | Default | 3.4 | Bayer et al. (2022) |
| | Max | 5.1 | Bayer et al. (2022) |
| | Min | 1.6 | Bayer et al. (2022) |

**Table 2.** Summary of the factors governing the relative abundances and water column rates of AOA and NOB in the dark, oxygenated ocean. The yield refers to the amount of biomass synthesized per amount dissolved inorganic nitrogen utilized.

| | Relative biomass yield | Relative biomass loss rate | Relative cell size (quota) | Relative uptake affinity | Reference |
|---|---|---|---|---|---|
| Relative biomasses | X | X | | | Eqn. C6 |
| Relative abundances | X | X | X | | Eqn. C7 |
| Relative nitrification rates | | | | | Eqn. C8 |
| Relative C fixation rates | X | | | | Eqn. C10 |
| $[\mathrm{NH_4^+}]:[\mathrm{NO_2^-}]$ | X | X | | X | Zakem et al. (2018) |

### 3.1.1 Relative abundances

The simple expressions capture the observed difference in the relative cellular abundances of AOA to NOB (Fig. 2a). The steady-state balances suggest that the relative cellular abundances of AOA to NOB reflect three factors: their relative biomass

yields, cell quotas, and population loss rates (Table 2, Eqn. C7). We can calculate the impact of each factor using our default parameter estimates to understand why the model captures the 7- to 11-fold observed difference. The higher biomass yield (2.3-fold on average) and smaller cell quota (3.5-fold on average) of AOA both contribute to the calculated higher abundance of AOA relative to NOB. In the default simple model (black line in Fig. 2a), we assume equal population loss rates. If AOA are subject to a higher mortality rate than NOB (as inferred by Zhang et al. (2020)), AOA abundance would be reduced and

the abundance difference would be smaller (though AOA abundance may still be higher than NOB abundance). If AOA have a lower mortality rate than NOB (as inferred by Kitzinger et al. (2020)), the abundance difference would be larger. However, we find that we do not need to invoke a difference in mortality rates to explain the observed relative abundances.

**Figure 2.** The simple expressions relating nitrifying microorganism abundances and rates compared to ocean observations and global model output below the 1% light level. a. and c. Abundances of ammonia-oxidizing archaea (AOA) and nitrite-oxidizing bacteria (NOB) (archaeal *amoA* and NOB 16S gene abundances from Zhang et al. (2020) and Santoro et al. (2010)). b. and d. Ammonia (NH$_3$) and nitrite (NO$_2^-$) oxidation rates (measured rates from Zhang et al. (2020); Dore and Karl (1996)).

.





### 3.1.2 Nitrification rates

The simple expressions capture the observed pattern of similar rates of $NH_3$ and $NO_2^-$ oxidation in the dark ocean (Fig. 2b; Dore
and Karl (1996); Ward et al. (2008); Zhang et al. (2020)). The model clarifies that at steady state, $NH_3$ and $NO_2^-$ oxidation rates
are both set solely by the $NH_4^+$ supply rate from heterotrophic excretion (Eqn. C8). Interestingly, this suggests that metabolic
and ecological traits of the nitrifying microorganisms do not impact the nitrification rates in steady-state, dark, oxygenated
environments (Table 2). Rather, population growth rates and abundances adjust to process the $NH_4^+$ supply at equal rates. In
contrast, in dynamic environments (when the steady-state approximation is not valid), nitrification rates may be decoupled
from $NH_4^+$ supply or from one another. Decoupling may also occur at the base of the euphotic zone where phytoplankton are
active, if either AOA or NOB is a better competitor than the other against phytoplankton for DIN. Additionally, $NO_2^-$ oxidation
may exceed $NH_3$ oxidation at steady state if other $NO_2^-$ is supplied, such as from anaerobic $NO_3^-$ reduction (Füssel et al., 2012;
Beman et al., 2013; Babbin et al., 2020; Santoro et al., 2021).

### 3.1.3 Carbon fixation rates

The simple expressions indicate that the relative carbon fixation rates of AOA and NOB in the water column are proportional
to their relative biomass yields (Eqn. C10). A thermodynamics-based theoretical estimate suggests that the AOA yield is
approximately 3-fold higher than the NOB yield when assuming equal cost of biomass synthesis (Zakem et al., 2018). Measured
biomass yields and direct measurements of C fixation rates relative to nitrification rates are consistent with the theoretical
estimates, with AOA yield 2–4 fold higher than NOB yield (Watson and Waterbury, 1971; Martens-Habbena et al., 2009;
Santoro and Casciotti, 2011; Spieck et al., 2014; Qin et al., 2014; Berg et al., 2015; Bayer et al., 2019; Zhang et al., 2020;
Kitzinger et al., 2020; Bayer et al., 2022). The recently published 2.3-fold higher yield value incorporated here suggests that
the cost of biomass synthesis may be higher for AOA than NOB (Bayer et al., 2022), which we discuss and analyze below.

### 3.1.4 No impact of uptake kinetics

At steady-state (i.e. $\frac{dB}{dt} \approx 0$), the parameters governing substrate uptake rates do not impact the nitrifiers' relative abundances
or rates. In contrast, previous work demonstrates how uptake kinetics (specifically, uptake affinities) do impact the steady-state
concentrations of $NH_4^+$ and $NO_2^-$ (Table 2; Zakem et al. (2018)). (To clarify, we note that in Zakem et al. (2018), the impact of
uptake affinity on cell abundance is due to an assumed correlation of affinity with cell size.) However, we expect that uptake
kinetics should impact nitrifier abundances and rates in dynamic environments and in the euphotic zone, when competition
with phytoplankton matters. For this reason, we turn to fully dynamic versions of the ecosystem model to determine to what
degree the simple, steady-state point balances are useful in interpreting nitrification rates and abundances at larger scales.

### 3.2 Vertical profiles

The water column ecosystem model captures much of the observed profiles from the W. Pacific Ocean (Fig. 3, Zhang et al.
(2020)). Nitrifier abundances and nitrification rates peak just below the euphotic zone, where remineralization rates are higher,





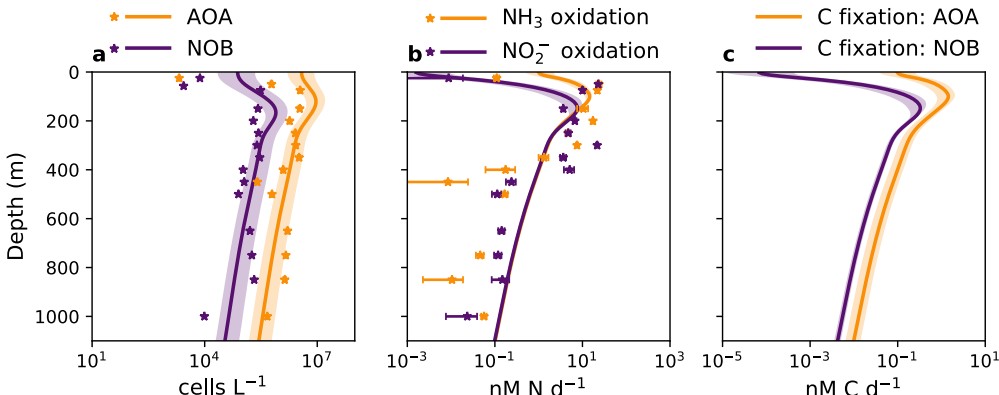

**Figure 3.** Water column model solutions compared to observations in the Western Pacific Ocean. Solid lines are model solutions and marked points are measurements from Zhang et al. (2020). The shaded areas denote the 5th and 95th percentiles of the ensemble of model solutions in which the AOA and NOB parameter values are varied. a. Abundances of ammonia-oxidizing archaea (AOA; measured with *amoA*) and nitrite-oxidizing bacteria (NOB; measured with 16S). b. $NH_3$ and $NO_2^-$ oxidation rates; c. Model prediction of the carbon fixation rates associated with AOA and NOB.

and then attenuate with depth as productivity declines. Below the surface, the model solutions capture the observed patterns and

converge to the simple expressions. AOA abundances exceed NOB abundances (Fig. 3a), and $NH_3$ and $NO_2^-$ oxidation rates decline in proportion with one another over depth (Fig. 3b). Since the associated *in situ* C fixation rates were not measured, we can use the model to predict the water column C fixation rates of AOA and NOB (Fig. 3c). AOA C fixation is significantly (2.3–fold) higher than NOB C fixation at depth, directly reflecting the higher yield of AOA.

The observations exhibit more variation in the deep ocean than the model solutions. Measured abundances and nitrification

rates increase and decrease together around the average state captured by the model. The lack of variability in the model reflects the simplifications of the one-dimensional physical configuration. In reality, time-varying circulation, vertical mixing, variability in the sinking organic matter flux, and biological patchiness can produce these fluctuations. To embrace some of this complexity, as well as other complexity due to variations in average surface productivity and oxygen availability, we next turn to the global model results.

**3.3 Global patterns and integrals**

We analyze nitrification activity at the global scale using the three-dimensional, global configuration of the ecosystem model. This allows us to investigate how the relationships among AOA and NOB vary across diverse environments and whether the simple expressions derived from the steady-state point balances are able to match the solutions given this complexity. We compare the global results to the simple expressions (Fig. 2c,d), and then calculate globally integrated nitrification and

associated C fixation rates.




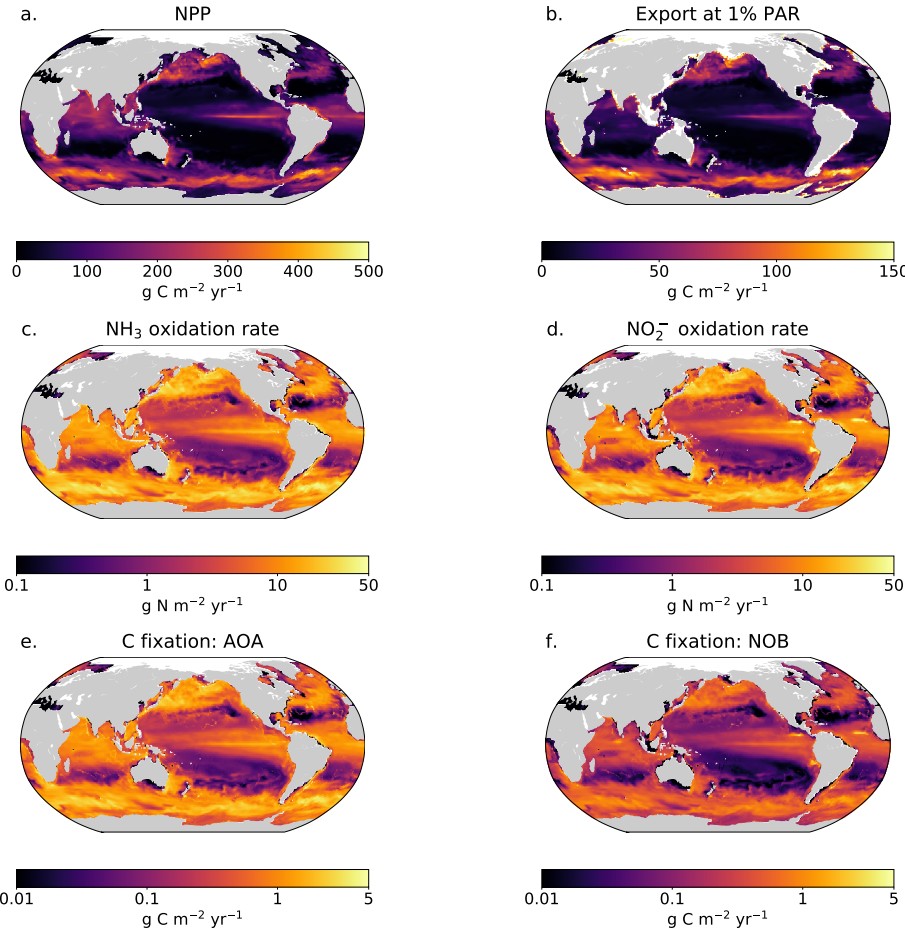

**Figure 4.** Output from the global ecosystem model (Darwin-MITgcm): a. Net primary productivity (NPP); b. Export of particulate organic carbon from the euphotic zone (calculated at the 1% light level). c. Ammonia oxidation rate; d. Nitrite oxidation rate; e. Rate of carbon fixation associated with ammonia-oxidizing archaea (AOA), f. Rate of carbon fixation associated with nitrite-oxidizing bacteria (NOB). All panels depict annual averages after 250 years of integration. Panels a and c–f depict vertically integrated quantities.

### 3.3.1 Correlation of NPP, export, and nitrification

The model demonstrates an expected global-scale correlation between NPP, the particulate organic carbon export flux, and nitrifier activity (Fig. 4), which is consistent with many observations (Ward et al., 2008; Newell et al., 2013; Smith et al., 2016; Santoro et al., 2017; Laperriere et al., 2020; Santoro et al., 2021). Vertically integrated nitrification rates and associated chemoautotrophic C fixation rates increase with NPP. Primary production indirectly fuels subsurface nitrification via the supply of sinking organic substrate and subsequent remineralization (Kirchman and Williams, 2000).





### 3.3.2 Deviations due to physical transport

Global model output matches the simple expressions to first order (Fig. 2c,d). However, there are significant deviations. Many of these deviations reflect the impacts of physical transport. Previous work has demonstrated how the physical transport of

biomass impacts microbial diversity and ecological interactions in locations where the timescales of transport are similar to the timescales of microbial growth (Clayton et al., 2013). Such short timescale physical events can result in the co-occurrence of nitrifiers and phytoplankton in locations in which one group would be otherwise outcompeted. Many of the shallower locations plotted in Fig. 2 are examples of this case.

### 3.3.3 Decoupling of nitrification rates

In many locations, AOA abundances and $NH_3$ oxidation rates are higher than NOB abundances and $NO_2^-$ oxidation rates (Fig. 2, c and d). In the model, AOA are better competitors than NOB against phytoplankton because of their higher biomass yield. Therefore, AOA can persist at higher light levels than NOB in the model. This may not be realistic, particularly if metabolically versatile NOB have an equally large or larger maximum potential growth rate than AOA (Kitzinger et al., 2020). Many of these events occur at or near the base of the euphotic zone, closer to the surface mixed layer, and so the transport of

nitrifier biomass upwards into the euphotic zone also contributes to this mechanism of decoupling.

  In contrast, NOB abundances and $NO_2^-$ oxidation rates are higher than AOA abundances and $NH_3$ oxidation rates at locations where $NO_2^-$ has accumulated due to anaerobic $NO_3^-$ reduction (Fig. 2, c and d; Fig. A1). The higher nitrification and C fixation rates of NOB (evident in Fig. 4 d and f) indicate the well-known locations of permanently anoxic oxygen minimum zones (Paulmier and Ruiz-Pino, 2009; Kwiecinski and Babbin, 2021). Enhanced $NO_2^-$ oxidation in and near anoxic zones is

consistent with observations (Füssel et al., 2012; Babbin et al., 2020; Saito et al., 2020; Santoro et al., 2021). Though both AOA and NOB growth becomes limited by oxygen supply in anoxic zones, physical transport and the accumulation of $NO_2^-$ (but not $NH_4^+$) enhances $NO_2^-$ oxidation rates on average in these areas.

### 3.3.4 Integrated rates

Modeled global NPP and nitrification rates are consistent with previous estimates (Table 3). Global NPP is similar to other

estimates at about 40 PgC yr$^{-1}$, equating to photoautotrophic N assimilation of about 7 PgN yr$^{-1}$. The modeled particulate organic matter export flux (12–13 Pg C yr$^{-1}$, or 2.1–2.4 Pg N yr$^{-1}$, at 1% PAR) is larger than other estimates (5–11 Pg C yr$^{-1}$; Schlitzer (2000); Henson et al. (2011); Siegel et al. (2014)). Global nitrification rates reach a magnitude that is a substantial fraction of N-based NPP, at 2–3 PgN yr$^{-1}$. Nitrification is higher than the export of organic nitrogen because nitrification emerges in the euphotic zone in the model: 10-30% of the nitrification rates are in waters at or above the 1% light level.

Despite the enhanced $NO_2^-$ oxidation near anoxic zones, the global $NO_2^-$ oxidation rate (2.1–3.0 PgN yr$^{-1}$) is lower than the global $NH_3$ oxidation rate (2.3–3.4 PgN yr$^{-1}$). This is due to the inferior competitive ability of modeled NOB relative to AOA against phytoplankton, as discussed above. These nitrification rates are within the range of other estimates of 1.5 – 4.6 PgN yr$^{-1}$ (Gruber, 2008; Wuchter et al., 2006).





**Table 3.** Globally integrated rates from the global ecosystem model (Darwin-MITgcm). Export is calculated as the flux of particulate organic matter at the base of the euphotic zone (the 1% light level). The range of nitrification and associated C fixation rates reflects sensitivity to the range of yield and population loss rate parameters of the modeled AOA and NOB functional type populations across multiple simulations.

| Globally integrated flux | Value | Units |
|---|---|---|
| **NPP** | | |
|     Carbon-based | 36–38 | Pg C yr$^{-1}$ |
|     Nitrogen-based | 6.5–6.7 | Pg N yr$^{-1}$ |
| **Organic export flux** | | |
|     Carbon-based | 12–13 | Pg C yr$^{-1}$ |
|     Nitrogen-based | 2.1–2.4 | Pg N yr$^{-1}$ |
| **NH$_3$ oxidation** | | |
|     Total | 2.3–3.4 | Pg N yr$^{-1}$ |
|     Euphotic zone | 0.3–1 | Pg N yr$^{-1}$ |
| **NO$_2^-$ oxidation** | | |
|     Total | 2.1–3.0 | Pg N yr$^{-1}$ |
|     Euphotic zone | 0.1–0.5 | Pg N yr$^{-1}$ |
| **Nitrifier C fixation** | | |
|     Total | 0.22–0.46 | Pg C yr$^{-1}$ |
|     AOA | 0.15–0.34 | Pg C yr$^{-1}$ |
|     NOB | 0.07–0.12 | Pg C yr$^{-1}$ |

Modeled global chemoautotrophic C fixation by the nitrifying populations is also within the range of previous estimates.

Modelled AOA fix $0.15 - 0.34 \, \text{PgC yr}^{-1}$ and NOB fix $0.07 - 0.12 \, \text{PgC yr}^{-1}$ (Table 3). Our point balance analysis allows us to determine the reason for the substantially higher C fixation values of AOA: the simple expressions clarify that this difference predominantly reflects the higher yield of AOA, and not the decoupling of nitrification rates. Summed together, the total model C fixation rate from nitrification is $0.22 - 0.46 \, \text{PgC yr}^{-1}$, about 1% of NPP (Table 3, Fig. 4). These values are similar to the 0.4 PgC yr$^{-1}$ estimated by Wuchter et al. (2006) for AOA, but nearly an order of magnitude less than the $\sim 1 \, \text{PgC yr}^{-1}$ estimated

by Pachiadaki et al. (2017) for NOB. For both AOA and NOB combined, the total is on par with the 0.40 PgC yr$^{-1}$ estimated by Middelburg (2011), higher than the 0.1–0.2 PgC yr$^{-1}$ estimated by Zhang et al. (2020), and higher than the 0.1 PgC yr$^{-1}$ estimated by Bayer et al. (2022). Though we use the same yields as Bayer et al. (2022), that study estimates a lower global chemoautotrophic C fixation rate than here because it incorporates a lower estimate of the organic export flux from the euphotic zone into the calculation and because the global model here also includes nitrification in the euphotic zone.

## 4 Discussion

### 4.1 Linking theory and observations

We develop quantitative relationships between AOA and NOB rates and abundances, derived from a theoretical model of the ecology of nitrification, that are consistent with observations in the open ocean. This alignment of theoretical and empirical understanding is a critical first step towards our ultimate goal of predicting how the nitrification ecosystem will change with





the environment. The relationships consist of simple, linear, yet mechanistic functions of a few metabolic and ecological parameters. Even with their simple forms, they serve to clarify the ecological dynamics at play in sometimes unintuitive ways. For example, it was not necessarily obvious that uptake kinetics should not influence nitrifier abundances or rates in the dark ocean, in contrast to dynamic (i.e. coastal or some surface) environments. Thus, this work has improved our understanding of the nitrification ecosystem.

### 4.2   Linking micro-scale and global-scale relationships


Our resulting relationships are relevant at the level of the cell as well as the level of the global marine ecosystem. For example, the measured yields of the nitrifying populations are important parameters for predicting global C fixation rates. This connection between the micro- and global-scales is one of the benefits of employing mechanistic microbial ecosystem models. In contrast, biogeochemical models that parameterize nitrification using a bulk rate constant do not provide the framework
necessary for directly linking laboratory measurements to global-scale dynamics. Furthermore, the mechanistic model allows for nitrification to emerge dynamically as a consequence of ecological interactions, rather than relying on prescribed light inhibition (Zakem et al., 2018). This results in significant rates of nitrification in the euphotic zone (10-30% of the global total).

### 4.3   Higher N yield but lower energetic efficiency of AOA

In this study, we employed recently published yields for AOA and NOB populations in environmentally relevant conditions for which the AOA yield is 2.3-fold higher than NOB (Bayer et al., 2022). This difference in yield is similar to the results of Kitzinger et al. (2020) (see discussion in Zakem et al. (2020)), but lower than the theoretical 3-fold difference estimated in Zakem et al. (2018). This suggests that marine NOB may be able to optimize their cellular machinery for greater overall efficiency of energy use than AOA. How can NOB obtain a higher energetic efficiency but a lower N yield? Using the metabolic
framework of Zakem et al. (2018), the fraction of electrons channeled towards synthesis vs. respiration (i.e. anabolism vs. catabolism) is represented as parameter $f$ (Rittman and McCarty, 2001). The yields can then be expressed as functions of $f$ for each type: $y_{NH_4} = 6d^{-1}f_{AOA}$ and $y_{NO_2} = 2d^{-1}f_{NOB}$. The coefficients reflect the elemental stoichiometry of the $e^-$-normalized redox reactions where biomass synthesis is normalized to one mol N. The ratio of the yields between AOA and NOB using this framework is $3f_{AOA} : f_{NOB}$. Therefore, $f_{AOA}$ can be smaller than $f_{NOB}$ while $y_{NH_4}$ is still larger than $y_{NO_2}$.
We can calculate $f$ using the yield values here ($y_{NH_4} = 0.098 \pm 0.021$ and $y_{NO_2} = 0.043 \pm 0.004$), the C:N of biomass used here (4.0 for AOA and 3.4 for NOB), and an estimate of $d$ ($d = 20 \pm 4$; Zakem et al. (2018)). This suggests the following $e^-$-partitioning fractions: $f_{AOA} = 0.08 \pm 0.02$ and $f_{NOB} = 0.13 \pm 0.03$. This suggests that AOA has a lower efficiency than NOB with respect to energy despite the significantly higher yield with respect to DIN uptake. This is also consistent with the calculations of Bayer et al. (2022).



### 4.4 An upper bound on chemoautotrophic C fixation from nitrification

The global ecosystem model is a useful tool for estimating global nitrification and associated C fixation rates. The model estimate integrates over the wide range in productivity rates across the ocean (Fig. 4), and allows for euphotic zone nitrification to emerge dynamically from microbial interactions. In the range of simulations used in this study, the contribution of nitrification to chemoautotrophic C fixation is $0.2 - 0.5$ PgC yr$^{-1}$. The contribution of AOA ($0.2 - 0.3$ PgC yr$^{-1}$) is higher than that of NOB (about 0.1 PgC yr$^{-1}$).

Despite the uncertainties inherent in global ecosystem models, we argue that this estimate constitutes a meaningful upper bound on the chemoautotrophic C fixation rates associated with nitrification. First, the nitrifier C fixation yields input into the model are higher than many previous estimates, and they include the fixed C that is lost to DOC release rather than just that incorporated into biomass (Bayer et al., 2022). Thus, our simulation provides an upper bound on the C fixation that is associated with a given nitrification rate. Second, the modeled export flux is larger than other estimates (Table 3, (Schlitzer, 2004; Henson et al., 2011; Siegel et al., 2014); section 2.2). Thus, the model should overestimate the "fuel" for nitrification at depth, which means that deep nitrification rates should also be overestimates. Third, our simulation includes a significant amount of emergent euphotic zone nitrification, and so 10-30% of nitrifier C fixation is within the euphotic zone in the model. Euphotic zone nitrification is widely observed (Ward, 1987; Dore and Karl, 1996; Ward, 2005; Stephens et al., 2020), though usually not accounted for in biogeochemical models that prescribe light inhibition for nitrification rather than allowing it to emerge from the interactions of dynamic nitrifying populations. For these reasons, the upper bounds of the modeled global C fixation rates (0.34 PgC yr$^{-1}$ for AOA and 0.12 PgC yr$^{-1}$ for NOB) are more likely to be overestimates than underestimates.

The upper bound of the modeled C fixation rate from nitrification (0.5 PgC yr$^{-1}$) is substantially less than the 1–10 PgC yr$^{-1}$ of deep carbon fixation estimated by Baltar and Herndl (2019). It is possible that metabolisms other than nitrification contribute to deep ocean C fixation. Given that nitrification rates wane sharply with depth in the dark ocean (Ward (1987); Dore and Karl (1996); Newell et al. (2013); Zhang et al. (2020); Figs. 2 and 3), the contribution of nitrification to deep C fixation may decrease substantially with depth (Pachiadaki et al., 2017). As nitrification wanes, a diverse microbial community carrying out other metabolisms may dominate C fixation rates (Swan et al., 2011).

### 4.5 Diagnosing additional metabolisms

The resulting quantitative relationships derived here can serve as a metric for determining whether additional chemoautotrophic C-fixing metabolisms or the metabolic versatility of NOB matter for large-scale biogeochemical cycling. Our upper bound global estimates suggest that the 10–fold higher rates of C fixation inferred in the deep ocean could be attributed to alternative (non-nitrifying) chemoautotrophic clades of microorganisms. Second, departures from the relationships in the deep ocean may be used to quantify departures from canonical nitrification. For example, recent studies suggest that a lifestyle of pure nitrification is not a valid assumption for many NOB because they exhibit metabolic versatility (Koch et al., 2014, 2015; Füssel et al., 2017; Bayer et al., 2020). Three factors impact the relative abundances of AOA to NOB for canonical nitrification (Eqn. C7): the relative yields, cell quotas, and loss rates (i.e. population turnover rates). If these three factors are constrained,





any additional difference in AOA:NOB may indicate an alternative metabolism at play. Therefore, careful measurements of the relevant parameters can help to tease apart these factors and quantify the contribution of alternative NOB metabolisms.

## 5 Conclusions and Outlook

This work provides simple, mechanistic relationships for the abundances and rates of AOA and NOB that are consistent with observations. The simple expressions explain multiple sets of *in situ* observations as linear functions of a few parameters. We provide an estimate of an upper bound on global carbon fixation rates from nitrification of 0.2–0.5 PgC yr$^{-1}$, with AOA contributing to higher rates than NOB. If dark ocean C fixation rates are higher than this upper bound, alternative C-fixing

metabolisms likely play a significant role in the marine carbon cycle.

*Code availability.* Water column model code, Darwin-MITgcm model code, and output files are available at Zenodo with the following DOI: https://doi.org/10.5281/zenodo.6384810.

**Appendix A: Model equations**

Here, we provide the full detailed set of equations used for the water column model. The three-dimensional ocean model

uses the same terms, but resolves additional phytoplankton and zooplankton functional types as well as the cycling of other elements, as explained above (section 2.2). All populations and nutrients are resolved as concentrations of nitrogen: the biomass of eight functional type populations (ammonia-oxidizing archaea $B_{AOA}$, nitrite-oxidizing bacteria $B_{NOB}$, two populations of phytoplankton $P_i$: a slower-growing, smaller, cyanobacteria-like gleaner and a faster-growing, larger, diatom-like opportunist (Dutkiewicz et al., 2009), heterotrophic bacteria $B_{het}$, and three microzooplankton grazers $Z_i$), three inorganic nutrients (NH$_4^+$,

NO$_2^-$, and NO$_3^-$), sinking particulate organic matter $POM$, and dissolved organic matter $DOM$. Nitrogen is conserved over the domain. Oxygen and micronutrients are assumed to be sufficiently abundant as to not limit the growth rates. All metabolic rates, including mortality rates, are modified as a function of temperature following the Arrhenius equation, following Dutkiewicz et al. (2015a) as outlined in Zakem et al. (2018). Each tracer $C$ is diffused by diffusion coefficients $\boldsymbol{\kappa}$ as:

$$\frac{\partial C}{\partial t} = \nabla \cdot (\boldsymbol{\kappa} \nabla C) + S_C \tag{A1}$$





where $S_C$ are additional sources and sinks as follows:

$$S_{B_{het}} = B_{het}(\mu_{het} - m_{lin_{het}} - m_Q B_{het} - gZ_3) \tag{A2}$$

$$S_{B_{AOA}} = B_{AOA}(\mu_{AOA} - \underbrace{m_{lin_N} - m_Q B_{AOA} - gZ_3}_{L_{AOA}}) \tag{A3}$$

$$S_{B_{NOB}} = B_{NOB}(\mu_{NOB} - \underbrace{m_{lin_N} - m_Q B_{NOB} - gZ_3}_{L_{NOB}}) \tag{A4}$$

$$S_{P_1} = P_1(\mu_{P_1} - m_{lin_{P_1}} - m_Q P_1 - gZ_1) \tag{A5}$$

$$S_{P_2} = P_2(\mu_{P_2} - m_{lin_{P_2}} - m_Q P_2 - gZ_2) \tag{A6}$$

$$S_{Z_1} = \zeta g Z_1 P_1 - m_Z Z_1^2 \tag{A7}$$

$$S_{Z_2} = \zeta g Z_2 P_2 - m_Z Z_2^2 \tag{A8}$$

$$S_{Z_3} = \zeta g Z_3 (B_{het} + B_{AOA} + B_{NOB}) - m_Z Z_3^2 \tag{A9}$$

$$S_{\mathrm{NH_4^+}} = -\frac{1}{y_{\mathrm{NH_4}}}\mu_{AOA}B_{AOA} - \mu_{NOB}B_{NOB} - V_{\mathrm{NH_4}}P + \underbrace{(\frac{1}{y_{het}}-1)\mu_{het}B_{het}}_{e_{NH_4}B_{het}} \tag{A10}$$

$$+ \underbrace{(1-\zeta)g\left[Z_1P_1 + Z_2P_2 + Z_3(B_{het}+B_{AOA}+B_{NOB})\right]}_{e_{NH_4}Z} \tag{A11}$$

$$S_{\mathrm{NO_2^-}} = (\frac{1}{y_{\mathrm{NH_4}}}-1)\mu_{AOA}B_{AOA} - \frac{1}{y_{\mathrm{NO_2}}}\mu_{NOB}B_{NOB} - V_{NO2}P \tag{A12}$$

$$S_{\mathrm{NO_3^-}} = \frac{1}{y_{\mathrm{NO_2}}}\mu_{NOB}B_{NOB} - V_{NO3}P \tag{A13}$$

$$S_{POM} = -\frac{1}{y_{het}}\mu_{het}B_{het}f_{POM} - \frac{\partial(w_s POM)}{\partial z} \tag{A14}$$

$$+ f_{mort}\left[\sum_i (m_{lin_i}B_i + m_Q B_i^2) + \sum_i (m_{lin_i}P_i + m_Q P_i^2) + \sum_i m_Z Z_i^2\right] \tag{A15}$$

$$S_{DOM} = -\frac{1}{y_{het}}\mu_{het}B_{het}(1 - f_{POM}) \tag{A16}$$

$$+ (1-f_{mort})\left[\sum_i (m_{lin_i}B_i + m_Q B_i^2) + \sum_i (m_{lin_i}P_i + m_Q P_i^2) + \sum_i m_Z Z_i^2\right] \tag{A17}$$

where $\mu_i$ is the growth rate of each microbial population calculated from the limiting uptake rate of the required substrates of

each. Heterotrophic bacteria growth is limited by the sum of DOM and POM according to uptake kinetic parameters (maximum

uptake rate $V_{max_{OM}}$ and half-saturation constant $K_{OM}$) and growth efficiency $y_{het}$. See Table A1 for all parameter values.

(Unless otherwise stated, parameters are identical to those in Zakem et al. (2018)). For phytoplankton, the growth rate is limited

by a maximum growth rate $\mu_{max}$, photosynthetic rate based on light availability, and the uptake $V_i$ of all three inorganic

nitrogen species as detailed in Zakem et al. (2018). Values for the maximum growth rate and the half-saturation constants



were computed as functions of cell size for a cell diameter (ESD) of 0.6 µm for P1 (cyanobacteria-like) and 20 µm for P2 (diatom-like) using the data-based allometric relationships in Litchman et al. (2007) as in Ward et al. (2012). The effective

half-saturation constants with respect to $\mu_{max}$ were calculated from those with respect to maximum uptake rate $V_{max}$ with an estimate of the minimum cell quota $Q_{min}$ from the relationships in Litchman et al. (2007), following Verdy et al. (2009) and Ward et al. (2012). Zooplankton populations grow at grazing rate $g$, calculated as a saturating function of their prey biomass with maximum grazing rate $g_{max}$, half-saturation constant $K_g$, and growth efficiency $\zeta$ (Armstrong, 1994; Zakem et al., 2018). $NH_4^+$ is excreted by heterotrophic bacteria and zooplankton in proportion to their growth efficiencies. DOM and POM are

sourced from the mortalities of all biomasses. $f_{POM}$ is the diagnostic fraction of total non-living organic matter in particulate form (i.e. POM/(POM + DOM)), and $f_{mort}$ is the assigned fraction of mortality that is partitioned into POM vs. DOM. POM sinks at rate $w_s$. In addition to grazing, microbial populations are subject to losses according to both linear mortality rate $m_{lin_i}$ and quadratic mortality rate $m_Q$, which represent losses to maintenance and senescence and losses to viral lysis, respectively. Quadratic mortality rate $m_Z$ represents predation of zooplankton by higher trophic levels.

**Appendix B:  Water column physical environment**

In the water column model, light and mixing attenuate with depth to form the stratified structure of a typical marine water column. Light energy $I$ decreases with depth $z$ according to the attenuation coefficients for water $k_w$:

$$I(z) = I_{in}e^{(-zk_w)} \tag{B1}$$

The mixed layer is imposed by varying the vertical diffusion coefficient $K_Z$ with depth, from a maximum $K_{max}$ at the surface

to a minimum $K_{min}$ with a length scale of $z_{mld}$. Vertical mixing increases at the bottom of the domain with a 100 m length scale, which avoids numerical error and simulates a bottom boundary mixed layer. $K_Z$ (m$^2$ s$^{-1}$) is calculated as:

$$K_Z = K_{max}e^{-\frac{z}{z_{mld}}} + K_{min} + K_{max}e^{-\frac{z-H}{100}} \tag{B2}$$

where H is the height of the domain (2000m).

**Appendix C:  Simple expressions**

We derive quantitative and mechanistic relationships between nitrifier abundances and rates that can be used to explain observations in the dark, oxygenated, open ocean. To accomplish this, we use a set of simplified model equations. We focus on the characteristics of the nitrification ecosystem below the sunlit layer, and so neglect phytoplankton activity. We neglect physical transport, since ocean transport rates are typically slow relative to microbial activity rates at depth. We assume that NOB, as well as AOA, consume either $NH_4^+$ or simple organic compounds such as urea for assimilation into biomass, but we neglect

this term in the equations because it is small relative to the other terms due to the low nitrifier yields (i.e. $y^{-1}\mu B >> \mu B$ using





the syntax explained below). This term is included in the full ecosystem model (including the 1D and 3D versions here), but it is negligible in all model solutions. With these simplifications, the following equations describe the relevant aspects of the nitrification ecosystem in the dark, oxygenated ocean:

$$\frac{dB_{AOA}}{dt} = B_{AOA}(\mu_{AOA} - L_{AOA}) \tag{C1}$$


$$\frac{dB_{NOB}}{dt} = B_{NOB}(\mu_{NOB} - L_{NOB}) \tag{C2}$$

$$\frac{d[\mathrm{NH_4^+}]}{dt} = e_{NH_4(B_{het}+Z)} - \underbrace{\frac{1}{y_{\mathrm{NH_4}}}\mu_{AOA}B_{AOA}}_{\text{NH}_3\text{ oxidation: uptake}} \tag{C3}$$

$$\frac{d[\mathrm{NO_2^-}]}{dt} = \underbrace{\frac{1}{y_{\mathrm{NH_4}}}\mu_{AOA}B_{AOA}}_{\text{NH}_3\text{ oxidation: excretion}} - \underbrace{\frac{1}{y_{\mathrm{NO_2}}}\mu_{NOB}B_{NOB}}_{\text{NO}_2\text{ oxidation: uptake}} \tag{C4}$$

$$\frac{d[\mathrm{DIC}]}{dt} = e_{\mathrm{DIC}(B_{het}+Z)} - \underbrace{\mu_{AOA}B_{AOA}}_{\text{C fixation: AOA}} - \underbrace{\mu_{NOB}B_{NOB}}_{\text{C fixation: NOB}}, \tag{C5}$$

where $B_{AOA}$ and $B_{NOB}$ (mol C L$^{-1}$) are the biomass concentrations of NH$_3$-oxidizing archaeal (AOA) and NO$_2^-$-oxidizing

bacterial (NOB) functional types, each with associated growth rate $\mu$ (t$^{-1}$), specific loss rate $L$ (t$^{-1}$), and yield $y$ (mol biomass synthesized per mol NH$_4^+$ or NO$_2^-$ utilized). The loss rate $L$ represents biomass losses to grazing, viral lysis, maintenance, and senescence. We include a simplified equation for dissolved inorganic carbon (DIC) to clarify how the model resolves nitrifier carbon fixation. Excretion of NH$_4^+$ ($e_{\mathrm{NH_4}}$) and DIC ($e_{\mathrm{DIC}}$) by heterotrophic biomass ($B_{het}$) represents the activity of both microheterotrophs and larger zooplankton.

We then analyze the steady-state balances of Eqns. 1–4. The steady-state approximation is valid when the changes in microbial biomass and nutrient concentrations are small relative to their fluxes (i.e. growth rates and nitrification rates), which captures the dynamics of the open ocean on average over time.

## C1 Relative abundances

Assuming steady state (e.g. $\mu_i = L_i$ and $\frac{d[\mathrm{NO_2^-}]}{dt} \approx 0$), we estimate the relative biomass concentrations $B$ (mol C L$^{-1}$) of AOA

and NOB from Eqns. C1, C2, and C4 as:

$$B_{AOA} : B_{NOB} = \frac{y_{\mathrm{NH_4}}}{y_{\mathrm{NO_2}}}\frac{L_{NOB}}{L_{AOA}} \tag{C6}$$

We can calculate cellular abundances $X$ (cells L$^{-1}$) using an estimate of the cell quota $Q$ (mol C cell$^{-1}$) as $X = BQ^{-1}$. This gives the relative abundances of AOA and NOB as:

$$X_{AOA} : X_{NOB} = \frac{y_{\mathrm{NH_4}}}{y_{\mathrm{NO_2}}}\frac{L_{NOB}}{L_{AOA}}\frac{Q_{NOB}}{Q_{AOA}} \tag{C7}$$





This suggests that the ratio of AOA to NOB cells is directly proportional to the ratio of their biomass yields and inversely proportional to the ratio of their loss rates and cell quotas.

## C2  Nitrification rates

The steady state of Eqns. C3 and C4 relates the three rates $r$ of DIN transformation (mol N L$^{-1}$ t$^{-1}$):

$$\underbrace{e_{NH_4(B_{het}+Z)}}_{r_{\text{NH}_4^+ \text{ supply}}} = \underbrace{\frac{1}{y_{\text{NH}_4}}\mu_{AOA}B_{AOA}}_{r_{\text{NH}_3 \text{ oxidation}}} = \underbrace{\frac{1}{y_{\text{NO}_2}}\mu_{NOB}B_{NOB}}_{r_{\text{NO}_2^- \text{ oxidation}}} \tag{C8}$$

This suggests that the three N-cycling rates – NH$_4^+$ supply from heterotrophic excretion, NH$_3$ oxidation, and NO$_2^-$ oxidation – are relatively equal in the dark ocean when other sources or sinks of NH$_4^+$ and NO$_2^-$ are negligible.

## C3  Carbon fixation rates

Assuming solely chemoautotrophic growth and no excess C fixation, the rate of carbon fixation is directly proportional to the production rate of each population $i$ as $C_{\text{fix}_i} = \mu_i B_i$, where $\mu$ is the growth rate and $B_i$ is the carbon-based concentration of

biomass. The relative C fixation rates (mol C L$^{-1}$ t$^{-1}$) for AOA and NOB are then:

$$C_{\text{fix}_{AOA}} : C_{\text{fix}_{NOB}} = \frac{\mu_{AOA}B_{AOA}}{\mu_{NOB}B_{NOB}} \tag{C9}$$

Plugging in the above expression for the relative biomasses (Eqn. C6) simplifies this ratio to:

$$C_{\text{fix}_{AOA}} : C_{\text{fix}_{NOB}} = \frac{y_{\text{NH}_4}}{y_{\text{NO}_2}} \tag{C10}$$

This suggests that the ratio of AOA carbon fixation to NOB carbon fixation is directly proportional to the ratio of their biomass

yields with respect to DIN utilization. Furthermore, the relationships between AOA and NOB C fixation rates and their respective nitrogen oxidation rates in the water column are:

$$C_{\text{fix}_{AOA}} : r_{\text{NH}_3 \text{ oxid.}} = y_{\text{NH}_4} \tag{C11}$$

$$C_{\text{fix}_{NOB}} : r_{\text{NO}_2^- \text{ oxid.}} = y_{\text{NO}_2} \tag{C12}$$

This suggests that at steady state, the macro-scale (water column) matches the micro-scale (cell): the ratio of the water column

C fixation rate to nitrification rate is directly proportional to the nitrifier's biomass yield with respect to DIN utilization. Excess C fixation, perhaps resulting in the excretion of DOC, decouples these relationships, since in this case the biomass yield (mol $B$ synthesized mol$^{-1}$ DIN) differs from the C fixation yield (mol C fixed mol$^{-1}$ DIN).





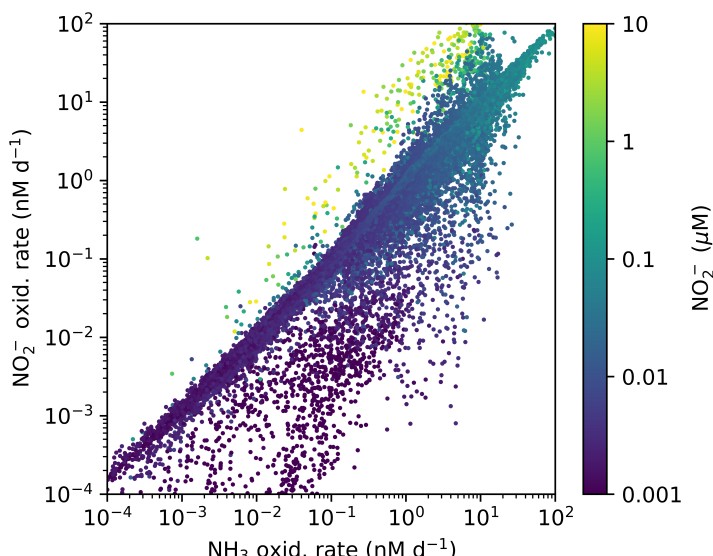

**Figure A1.** Annually averaged aerobic $NO_2^-$ oxidation rate against $NH_3$ oxidation rate with associated local concentration of $NO_2^-$ in the global ecosystem model (Darwin-MITgcm). The locations where $NO_2^-$ oxidation exceeds $NH_3$ oxidation correspond to the anoxic zone locations where $NO_2^-$ has accumulated due to anaerobic $NO_3^-$ reduction. Physical transport combines $NO_2^-$ and $O_2$. See Zakem et al. (2019) for detail of anaerobic functional type parameterizations.





**Table A1.** Parameter values for the water column model, including ranges for the ensemble.

| Parameter | Symbol | Value | Range | Units | Note |
|---|---|---|---|---|---|
| **Nitrifying populations:** | | | | | |
| $NH_4^+$ yield, AOA | $y_{NH_4}$ | $0.098 \pm 0.021$ | Gaussian | mol C (mol $NH_3$)$^{-1}$ | * |
| $NO_2^-$ yield, NOA | $y_{NO_2}$ | $0.043 \pm 0.004$ | Gaussian | mol C (mol $NO_2^-$)$^{-1}$ | ** |
| AOA cell quota | $Q_{AOA}$ | 11.5 | 10.8–14.9 | fg C cell$^{-1}$ | † |
| NOB cell quota | $Q_{NOB}$ | 39.7 | 23.8–54.7 | fg C cell$^{-1}$ | †† |
| Maximum DIN uptake rate | $V_{maxN}$ | $(50.8 \pm 4.68)$ | Gaussian | mol N mol N$^{-1}$ d$^{-1}$ | § |
| DIN half-saturation | $K_N$ | $133 \pm 38$ | Gaussian | nM N | § |
| AOA biomass C:N | $R_{NC_{AOA}}$ | 4.0 | | unitless | † |
| NOB biomass C:N | $R_{NC_{NOB}}$ | 3.4 | | unitless | †† |
| Linear mortality rate, nitrifier | $m_{lin_N}$ | 0.1 | 0.05–0.15 | d$^{-1}$ | §§ |
| **Heterotrophic bacteria and OM:** | | | | | |
| Maximum OM uptake rate | $V_{maxOM}$ | 1 | | mol N mol N$^{-1}$ d$^{-1}$ | |
| OM half-saturation | $K_{OM}$ | 0.1 | | µM N | |
| Yield, $B_{het}$ | $y_{het}$ | 0.14 | | mol N mol N$^{-1}$ | |
| Linear mortality rate, $B_{het}$ | $m_{lin_{Bhet}}$ | $0.02 (= 0.15\mu_{max})$ | | d$^{-1}$ | §§ |
| Fraction of mortality to POM vs DOM | $f_{mort}$ | 0.5 | | unitless | |
| **Phytoplankton growth:** | | | | | |
| Maximum growth rate, $P_1$ | $\mu_{max}$ | 0.515 | | d$^{-1}$ | ‡ |
| Maximum growth rate, $P_2$ | $\mu_{max}$ | 3 | | d$^{-1}$ | |
| $NO_x^-$ half-saturation, $P_1$ | $K_{NO_x P_1}$ | 0.0036 | | µM | ‡ |
| $NO_x^-$ half-saturation, $P_2$ | $K_{NO_x P_2}$ | 0.33 | | µM | ‡ |
| $NH_4^+$ half-saturation, $P_i$ | $K_{NH_4 P_i}$ | $0.5 K_{NO_x P_i}$ | | nM | ‡ |
| Linear mortality rate, $P_1$ | $m_{lin_{P1}}$ | $0.077 (= 0.15\mu_{max})$ | | d$^{-1}$ | §§ |
| Linear mortality rate, $P_2$ | $m_{lin_{P2}}$ | $0.45 (= 0.15\mu_{max})$ | | d$^{-1}$ | §§ |
| Chl $a$ absorption, $P_1$ | $a_{phy}^{chl}$ | 0.04 | | m$^2$ (mgChl)$^{-1}$ | # |
| Chl $a$ absorption, $P_2$ | $a_{phy}^{chl}$ | 0.01 | | m$^2$ (mgChl)$^{-1}$ | # |
| **Grazing and other mortality:** | | | | | |
| Maximum grazing rate | $g_{max}$ | 2 | | d$^{-1}$ | |
| Grazing half-saturation | $K_g$ | 1 | | µM N | |
| Grazing efficiency | $\zeta$ | 0.5 | | unitless | |
| Quadratic mortality rate, microbial | $m_Q$ | 0.1 | | µM N$^{-1}$ d$^{-1}$ | §§ |
| Quadratic mortality rate, $Z_i$ | $m_Z$ | 0.7 | | µM N$^{-1}$ d$^{-1}$ | §§ |
| **Physical parameters:** | | | | | |
| Maximum incoming PAR flux | $I_{max}$ | 1400 | | W m$^{-2}$ | |
| PAR attenuation in water | $k_w$ | 0.04 | | m$^{-1}$ | |
| Mixed-layer attenuation depth | $z_{ML}$ | 20 | | m | |
| Minimum vertical mixing coefficient | $K_{min}$ | $1 \cdot 10^{-4}$ | | m$^2$ s$^{-1}$ | |
| Maximum vertical mixing coefficient | $K_{max}$ | $10^{-2}$ | | m$^2$ s$^{-1}$ | |
| POM sinking rate | $w_s$ | 10 | | m d$^{-1}$ | |

*Table 2 in Bayer et al. (2022) for *Ca.* Nitrosopelagicus brevis U25 and *Nitrosopumilus sp.* CCS1 (natural seawater).

**Table 2 in Bayer et al. (2022) for *Nitrospina gracilis* Nb-3 (natural seawater).

†Table 1 in Bayer et al. (2022) for *Ca.* Nitrosopelagicus brevis U25 and *Nitrosopumilus sp.* CCS1 (all growth stages).

††Table 1 in Bayer et al. (2022) for *Nitrospina gracilis* Nb-3 (all growth stages).

§ From Martens-Habbena et al. (2009) for AOA, with conversion to N-based biomass as in Zakem et al. (2018).

§§Mortality rates, like all metabolic rates, are modified by temperature as in Zakem et al. (2018).

‡Computed using data-based allometric relationships in Litchman et al. (2007) as in Ward et al. (2012).

#Following Dutkiewicz et al. (2015a). See Zakem et al. (2018) for photosynthesis rate parameterization.





*Author contributions.* EJZ designed and executed the research. BB, AS, WQ, and YZ contributed to the model development. NML contributed to the analysis of the results. EJZ wrote the paper with contributions from BB, AS, YZ, and NML.

*Competing interests.* The authors declare that they have no conflict of interest.

*Acknowledgements.* EJZ was supported by the Simons Foundation Postdoctoral Fellowship in Marine Microbial Ecology. BB was supported by the Austrian Science Fund (FWF) project J4426-B. AES was supported by a Simons Foundation Early Career Investigator Award in Marine Microbial Ecology and Evolution (#345889) and the US National Science Foundation (Award OCE-1924512). YZ was supported by the National Science Fund for Distinguished Young Scholars (#42125603). NML was supported by the Simons Foundation: The Simons

Collaboration on Principles of Microbial Ecology (PriME #542389).





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
