# Peer review of "Controls on the relative abundances and rates of nitrifying microorganisms in the ocean"

_Biogeosciences, 2022_

## Author Response (AR1)

Comment 1

Zakem et al set out to evaluate the global contribution of nitrification to global N and C cycles. The approach is to apply a previously developed ecosystem model (Zakem et al. 2018) that resolves growth, respiration and loss rates of ammonia- and nitrite-oxidizers (AOA and NOB), as well as several other important biological and inorganic nutrient components. The new addition to parameterization of the model is the recently published (Bayer et al. 2022) information on cellular C and N quotas and yields for AOA and NOB.

Nitrification rates in the model are driven by the release of NH4 from remineralization of organic matter. It is stated that the remineralization flux in this model is larger than that produced by other models (L149), without explaining why that is so. It may be explained in the previous paper (does it result from the heterotrophic parameterization of Zakem et al 2018 and if so, how?), but it would be good to explain that briefly here, as this dependence on remineralization is fundamental to the outcome of the exercise.

Thank you for these points. In the revised version, we will explain how the upper estimate of the remineralization flux at depth is due to the export flux, which yes, is a consequence of the heterotrophic parameterization originating from the 2018 publication. The parameters that dictate the rate of organic matter uptake by heterotrophic microbes combined with the sinking rate of the POM control the export flux. This parameterization is (necessarily) simple in our model, and the parameter values are uncertain, and so we chose values that gave this upper estimate of export. The magnitude matters too: there is a very wide range in the data-based and model-based estimates of the export flux (from 5-11 PgC/yr), and ours is 12-13 PgC/yr. We intentionally embraced this overestimate so that we could robustly conclude an upper estimate on global nitrification and C fixation rates. However, as you point out, this needs to be thoroughly explained in the text.

This paragraph has been modified as:

**"Additional uncertainty in global nitrification and associated carbon fixation rates exists due to uncertainty in the flux of organic matter exported out of the sunlit surface. As we later clarify in our results, nitrification rates at depth are predominantly set by this export flux, via the supply of $NH^+_4$ from its remineralization. Therefore, we treat the uncertainty due to the export flux by considering that the export flux in our global model is larger ($12–13 \text{ Pg C yr}^{-1}$) than previous estimates ($5–11 \text{ Pg C yr}^{-1}$; Schlitzer (2000); Henson et al. (2011); Siegel et al. (2014)). In the model, the export flux is controlled by the parameters that dictate the rate of organic matter uptake by heterotrophic microbes combined with the sinking rate of the particulate organic matter (POM). Here, we use the same heterotrophic activity parameterization as in Zakem et al. (2018). Because the model provides an upper bound on the export flux, it provides an upper bound on deep nitrification rates. This is a pragmatic approach because the resulting modeled global rates are lower than many previous estimates. The resulting range of the export flux in Table 3 reflects the choice of cutoff to exclude the very high values of export in the coastal grid points, where the model has no skill. Across the three global model versions, the export flux remained the same with respect to the degree of accuracy represented in Table 3."**

Despite this larger remineralization flux, it is found that total nitrification is on the low end of estimates obtained from other sorts of models. The authors argue that their numbers are reasonable and better, because not only are the other outputs from their model reasonable, but the new quota and yield parameterizations are both realistic and data based. Their higher remineralization flux would have had

the opposite effect, implying that real physiology of the microbes is responsible. How much lower would the nitrification rates have been at the lower remineralization rates of other models?

Thank you: the answers to this question will be helpful to flesh out in the text. To first order, the nitrification rates at depth correlate with the exported remineralization flux in a linear fashion. Nitrification within the euphotic zone then adds to give total nitrification. In Table 3, subtracting the euphotic zone nitrification from the total matches the N-based export shows this relationship (though some nonlinear interactions between the dynamic biomass populations make it not precisely so). We will make this more explicit in Table 3 by adding lines with "Below euphotic zone" totals for NH3 and NO2 oxidation, which will roughly match the N-based organic carbon flux total above. Also, we will explain within the text that should the export flux be different, the "Below euphotic zone" nitrification totals will scale with this export, and so the total nitrification rates can be anticipated in this way. For example, if our model had a C-based export flux of 10 PgC/yr (equating to roughly 1.5 PgN/yr) then our below-euphotic zone nitrification rates would be roughly 1.5 PgN/yr instead of the 2.0-2.5 that they are now. If our C-based export was 5 PgC/yr (equating to roughly 0.75 PgN/yr) then our below-euphotic zone nitrification rates would be roughly 0.75 PgN/yr.

In the revised version, we have added the "Dark" nitrification rates to Table 3. Dark NH3 oxidation ranges from 2.0 to 2.3 PgN/yr, and dark NO2 oxidation ranges from 1.9 to 2.1 PgN/yr. These are the rates below the 1% light level at all locations.

We also have added the following in the Results section:

**"Due to our heterotrophic activity parameterization, the modeled particulate organic matter export flux (12–13 Pg C yr$^{-1}$, or 2.1–2.4 Pg N yr$^{-1}$, at 1% PAR) is larger than other estimates (5–11 Pg C yr$^{-1}$; Schlitzer (2000); Henson et al. (2011); Siegel et al. (2014)). This choice to overestimate the export flux allows us to provide a meaningful constraint on global rates. Because deep nitrification rates are set by the export flux of organic nitrogen, the upper bound on the export flux provides an upper bound on dark nitrification rates. Variation in nitrifier parameter values does not significantly change the export flux. In all models, the export flux remained the same with respect to the degree of accuracy represented in Table 3."**

**"Dark nitrification rates are, as expected, roughly equal to the export flux of organic nitrogen (Table 3). Thus, a lower export flux would lower modeled dark nitrification rates proportionally."**

The underlying model has been published before and my expertise does not equip me to critique it carefully, so I will take it as acceptable and go from there to comment on a few other aspects of the work. I found the paper very clearly written and very readable, logically developed without redundancy. The main points were clear and generally well supported and linked directly to the calculations.

This is wonderful to hear.

The authors emphasize some of the major outcomes of their model, which I agree are interesting and important, but perhaps not quite as novel as they imply.

Yes, we are still struggling to highlight what we think is actually new and helpful to the community: how the differences in AOA vs NOB physiology translate into observed differences in the water column and differences in global totals. For example, the relative loss rates of AOA vs NOB have been inferred

differently by different recent papers (Zhang et al 2020 and Kitzinger et al 2020). Kitzinger et al. (2020) inferred that NOB loss rates must be higher than those of AOA to explain their relative abundances. We show that because of the control by the other parameters, we don't need to invoke a difference in loss rates. The reconciliation lies in the delineation of energetic vs. DIN yield (which we explain in our paragraph beginning on line 320). Additionally, rather than just providing a number for global nitrification and associated C fixation, our analysis allows us to interpret those results and robustly explain *why* the model produces the numbers that it does: AOA C fixation exceeds NOB C fixation mainly because of the differences in DIN yield. We will try to highlight these contributions more clearly in the revised version. In the abstract, we modified the following sentence to emphasize this: **"Modeled carbon fixation by AOA (0.2–0.3 Pg C yr$^{-1}$) exceeds that of NOB (about 0.1 Pg C yr$^{-1}$) because of the significantly higher DIN yield of AOA, despite its lower energetic efficiency."**

-The finding that nitrification in the euphotic zone comprises up to 30% of the global total:  It would be good to mention and cite Yool et al (2007) as an earlier model (which was based on a lot of actual rate measurements) that did indeed consider nitrification in the euphotic zone and found that it was very significant, providing substantial recycled NO3 to support primary production.

Yes, thank you. It was an oversight to not reference Yool et al 2007. We have now included this reference as: **"Furthermore, because the mechanistic model allows for nitrification to emerge dynamically, rather than relying on prescribed light inhibition, the model anticipates signif- icant rates of nitrification in the euphotic zone (10-30% of the global total). As Yool et al. (2007) articulate, this impacts the relationship between nitrate and "new production" in the ocean."**

-Uptake kinetic parameters are not important in determining abundances or rates in the deep ocean:  That is an interesting finding, but the inverse, which they state, is even more interesting – that kinetics are important in more dynamic settings. Since the upper ocean (bottom of the photic zone) is where nitrification rates are highest, and kinetics are important there, then kinetics are important in the overall picture.  Others have published plenty of data showing lack of relationship between in situ substrate concentrations and measured rates (which implies that substrate concentration is not the controlling factor). One small data set which directly supports the contention of Zakem et al here is the paper on nitrite oxidation by Sun et al (2017).  They measured substrate kinetics and found that correcting for substrate affinity did not affect apparent rates below the surface layer.

Yes, the inverse is indeed interesting. One implication is that the nitrification rates in the euphotic zone are less certain than the below surface rates. Thank you for making this point. We will include this in the revised version of the paper, and also cite Sun et al 2017.

We now include the following text:

**"For example, it was not necessarily obvious that uptake kinetics should not influence nitrifier abundances or rates in the dark ocean, in contrast to dynamic (i.e. coastal or some surface) environments, where population differences in uptake kinetics would matter. This is consistent with the conclusions of Sun et al. (2017) that differences in substrate affinity do not impact the apparent rates of NO$^-_2$ oxidation below the surface layer."**

L 279: I suggest actually citing a paper for the previous estimates of Global NPP.  Maybe something like Anav et al. 2013 (J Climate), which has a figure showing a lot of different model estimates.

Thank you for the reference. We've cited Séférian et al. 2020 (Current Climate Change Reports) which has updated model results for CMIP6 (updated from the CMIP5 results described in Anav et al. 2013) and also compares these with satellite-based estimates within the text. The text now reads: **"Global NPP is similar to other global model and remote-sensing-based estimates at about 40 Pg C yr$^{-1}$ (Séférian et al., 2020)".**

Several places in the text: I think they have the wrong Ward (2008) citation in the reference list. I don't know why they would be citing a paper about copper limitation of denitrification here.

Yes, thank you, we meant a different Ward 2008. We have replaced it with: Ward, B. B.: Nitrification in marine systems, in: Nitrogen in the Marine Environment.

Comment 2

This study by Zakem et al. uses a global microbial ecosystem model to estimate controls, rates, and abundances of nitrifying microbes (AOA and NOB) in the ocean. Their microbial ecosystem model is based on characteristics of known AOA and NOB communities, which allows predictions of their abundances and rates to emerge in a dynamically consistent way without having to prescribe simple rate functions like most global biogeochemical models. There still seems to be considerable uncertainty in some parameters, which was addressed with an ensemble of model simulations. They use measurements on rates and yields to distinguish the different parameters between AOA and NOB functional types to best estimate their abundances and rates, using three approaches starting with a steady-state 0D model to validate the core microbial model, then with a vertical water column, and finally with a global 3D model. They find that the NH3 and NO2 oxidation rates are mostly consistent in the deep, oxygenated ocean and primarily driven by the export of organic matter to the local system. Global NO2 oxidation rates are slightly lower than NH4 oxidation due to their model predicting NOB are less competitive against phytoplankton relative to AOA. An important finding is that AOA fixes about twice as much carbon mainly due to their higher yield compared to NOB. Their model estimates a global carbon fixation rate of 0.2-0.5 Pg C yr-1 which is a small fraction of global net primary productivity.

Overall I find this to be an important and informative study on global nitrifying microbial communities and their associated rates. I think it was very well written with an ideal balance between a concise technical description and understandable results. The model results and caveats are fairly addressed and discussed. My only minor criticism is that some additional insights and discussion could be provided in the paper (see minor comments below).

Best regards,

Chris Somes

GEOMAR Helmholtz Centre for Ocean Research Kiel

Thank you very much for your careful analysis and constructive feedback, Chris.

Minor Comments:

Figure 4: NPP and Export patterns

It is interesting to me that your global NPP rates are consistent with most estimates whereas the export is on the very high-end of the estimates. I wonder if that has something to do with relatively high nitrification occurring in the euphotic zone.

Good question. In our model, NPP and the export flux are able to be decoupled because of our dynamic remineralization. The decoupling is predominantly controlled by the parameterization of heterotrophic microbes that consume POM and the sinking rate of the POM. So, to first order, nitrification in the euphotic zone does not control this decoupling. Rather, euphotic zone nitrification can be thought of as, predominantly, the "gleaning" of reduced DIN that phytoplankton are otherwise unable to assimilate, mostly when they are partially light-limited towards the base of the euphotic zone. In some dynamic areas, bloom-like conditions can result in abundant DIN supply in the euphotic zone in which case nitrification may coexist with phytoplankton because competitive exclusion has not yet taken place, but this would have a smaller impact on NPP from the fact that phytoplankton would have access to more NO3 than NH4, which has a small energetic cost (reflected in slightly lower uptake kinetics by

phytoplankton in the model). In the revised manuscript, we will better explain the controls on the export flux, their uncertainties, and the choices that we make about them in the parameter values.

In the Methods, we add the following:

**"In the model, the export flux is controlled by the parameters that dictate the rate of organic matter uptake by heterotrophic microbes combined with the sinking rate of the particulate organic matter (POM)."**

In the Results section, we now include the following:

**"Because deep nitrification rates are set by the export flux of organic nitrogen, the upper bound on the export flux provides an upper bound on dark nitrification rates. Variation in nitrifier parameter values does not significantly change the export flux. In all models, the export flux remained the same with respect to the degree of accuracy represented in Table 3."**

**"Dark nitrification rates are, as expected, roughly equal to the export flux of organic nitrogen (Table 3). Thus, a lower export flux would lower modeled dark nitrification rates proportionally."**

I'm surprised to see highest NPP and export rates in the Southern Ocean on the annual average, is that consistent with other estimates? If export is overestimated in the Southern Ocean, would that imply NPP might be underestimated in the low latitudes? Does export efficiency (including through the twilight zone) change significantly between low and high latitudes which could alter vertical profile total nitrification rates in different regions?

The high NPP in the S. Ocean is consistent with other published Darwin model estimates. The observations are fairly uncertain there. Stephanie Dukiewicz's work comparing the Darwin ecosystem model output to observed NPP and chlorophyll does suggest: 1. Observations, when they are there, suggest lower productivity in the S. Ocean, and 2. Observations are often missing in that area. (See, for example, Dutkiewicz et al. 2015 *Biogeosci.* Fig. 6 and Dukiewicz et al. 2019 *Nat. Comm* Fig. 1 a and c). Conversely, as you suggest, the model predicts lower productivity in the oligotrophic gyres than observations suggest (see same references/figures). Since export scales with NPP to first order (our Fig. 4b), yes, it changes significantly along latitude following this pattern and yes, it likely has this same bias when compared to observations (though this is speculation, since our global observations of export are limited). And similarly, nitrification rates could be biased high at high latitudes and low at low latitudes. It would be interesting to do a spatial analysis diving into this, and work on fixing the ever-present problem of how to spread sufficient nutrients into the gyre centers in order to support higher productivity rates there. However, for the purposes of this study, we think the global integrals are still useful estimates. We will include discussion of this model bias, and the uncertainty in S. Oc. NPP in both model and observations, in the revised manuscript.

We now include the following statement:

**"Like previous comparisons of Darwin-MITgcm model simulations with satellite-based observations of NPP (Dutkiewicz et al., 2015b, 2019), modeled NPP is higher in the S. Ocean and lower in oligotrophic gyres than observations suggest, though observations are sparse at high latitudes. The modeled export flux is closely coupled with modeled NPP, and so the export flux likely contains similar biases."**

Lines 317-318: 10-30% of global total

It is intriguing to me that your analysis suggest up to 30% of global nitrification may occur in the euphotic zone. In Figure 3b, it even appears your model is significantly underestimating NO2 oxidation at the base of the euphotic zone. I'm curious about this uncertainty range as I see very little shading around the model lines in Figure 3b.

Thank you for pointing this out. Yes, we realize that our results seem to suggest that there is both huge uncertainty (10-30% of euphotic zone nitrification in 3D) yet very little uncertainty (in 1D). In both models, we vary only the parameters that directly influence the growth and mortality of the nitrifiers (as described in Methods). Because of the dynamic environments in the 3D model, in contrast to the strictly steady state solutions found in 1D, the uncertainty in these parameters results in a much wider range of realized solution space in 3D. However, there is much more uncertainty in the model from other parameters, both in the 1D and 3D configurations. For example, in neither model do we vary the parameters that impact the export flux (i.e. heterotrophic bacteria) or the physical environment, and these parameters would change the depths and intensities of nitrification in both 1D and 3D. Therefore, the mismatch between model and data in the 1D model (Fig. 3) is likely controlled predominantly by this uncertainty that we do not explicitly consider in the ensemble. When we do incorporate this additional uncertainty, the shaded areas become large everywhere, and we lose what we think now is a helpful aspect of interpretability: the fact that abundances and C fixation rates reflect the uncertainty from parameter variations but the nitrification rates remain robust because they are controlled by the export flux. For similar reasons, we did not want to consider all of the uncertainty in all of the ecosystem parameters in the 3D configuration. Also, dealing meaningfully with the full model uncertainty in 3D requires a very involved approach (i.e. 3D ensemble with variation in hundreds of parameters, Monte-Carlo style), which the Darwin-MITgcm configuration has not been designed to do. Thus we took the approach of here relying on an upper estimate of the export flux in order to make the conclusions that we do.

We have modified the following statement as:

**"However, there are significant deviations compared to the equilibrium solutions in the water column model. Many of these deviations reflect the impacts of physical transport."**

Lines 271-272: "NOB … are higher than AOA … due to anaerobic NO3 reduction"

I find it interesting that the highest NO2 oxidation rates in the global ocean occur near oxygen deficient zones. I wonder how well ODZs are reproduced and how that factors into the uncertainty given the very high rates (I think you mean Fig. 4 c and d instead of Fig. 2 here since I don't see any indication of oxygen in Fig. 2). For example, I don't see any hot spot in the Arabian Sea ODZ and there appears to be a hot spot off the North African Eastern Boundary Upwelling System that is not related to export which is typically not anaerobic.

Yes, we consider the ODZs as only qualitatively reproduced, in that anoxic zones form in roughly the right areas (though as you say, missing the Arabian Sea ODZ and having too much O2 depletion in the S. Atlantic). We did not analyze their extent in the model or the mismatch between observations and model. You are correct that this could add uncertainty to our global totals. It would be straightforward and clarifying to calculate the amount of NO2 oxidation in the model ODZs and include that in the analysis in the revised manuscript. Since global NO2 oxidation is lower than NH4 oxidation, we can tell that it has a smaller effect than the higher competitive ability of AOA vs. NOB relative to phytoplankton in the model, but it will still be interesting to quantify. Thank you for this suggestion. Regarding the figures: Fig. 4 is indeed the right reference for this statement, thank you.

We have added the following sentence, which quantifies that the contribution to global nitrification rates in anoxic regions is small:

**"Where $[NO^{-}_{2}] > 10$ μM in the annually 275 averaged solutions, integrated $NO^{-}_{2}$ oxidation is roughly 10x higher than $NH_3$ oxidation (about 20 vs 2 Tg N $yr^{-1}$)."**

Section 4.2:

Most global biogeochemical models estimate nitrification based on the amount of particulate organic matter (from export) that remineralizes in each location, which you also acknowledge (lines 147-148) is the main driver of nitrification rates in your model. Thus, I am not completely convinced that global biogeochemical models that do resolve microbial ecosystem functional types cannot provide reliable estimates on global nitrification rates, so perhaps you can be more specific about what you mean by "biogeochemical models that parameterize nitrification using a bulk rate constant do not provide the framework necessary for directly linking laboratory measurements to global-scale dynamics".

It is true that models with implicit nitrification should in principle be able to estimate deep nitrification rates accurately, if the export flux is estimated accurately. This follows from the conclusions that we make about the steady state nitrification rates and their insensitivity to nitrifier parameters, and we will emphasize this in the text. What we meant by this statement is that the bulk rate constants used in implicit schemes cannot be constrained by the measured values that we use to describe AOA and NOB in the model here. At least, that is our understanding. In contrast, we can use the measured values directly into our parameterization. Therefore, the models with implicit nitrification can get the rates right (if our analysis here is correct that the kinetic parameters don't matter!), but they don't help us connect the dots between nitrification rates and associated abundances of organisms, for example, that we would need to start making connections with sequencing datasets. We will revise the statement here to clarify what we mean and emphasize that our results actually support the ability of implicit nitrification schemes to predict nitrification rates.

We have changed this section so that it states only that this type of model is able to capture euphotic zone nitrification, without implying that nitrification is better estimated at depth (in the dark):

**"Furthermore, because the mechanistic model allows for nitrification to emerge dynamically, rather than relying on prescribed light inhibition, the model anticipates signif- icant rates of nitrification in the euphotic zone (10-30% of the global total). As Yool et al. (2007) articulate, this impacts the relationship between nitrate and "new production" in the ocean."**

One important exception is nitrification occurring in the euphotic zone. If possible, perhaps you can provide some insights or recommendations about how global biogeochemical models unable to explicitly resolve microbial functional types could best parameterize this process?

Interesting question. Perhaps there is a way of considering when "excess" reduced DIN exists in the euphotic zone. Primary production (or, phytoplankton growth, if phytoplankton are resolved) could be calculated in two ways: 1. Calculating the potential primary production that would occur if it were only limited by DIN, and 2. Calculating it way it is already being calculated, taking into account limitation by DIN, light, and other nutrients. Then, you could subtract 2 from 1 to give a rate of potential DIN assimilation that isn't being reached because phytoplankton are limited by something else. You would need to take care and also assume that nitrifiers require other nutrients (such as Fe), though at different ratios than phytoplankton relative to DIN uptake. You could then have an implicit rate of euphotic zone

nitrification. I think this might work, but it would need to be worked out in a model comparison, and ideally constrained by observations as well. Our euphotic zone estimations still need to be tested with data. We should collaborate!

Section 4.4: "first" (lines 342-344) and "third" (lines 347-350) reasons

These appear to be processes that are more realistically accounted for in your model estimate compared to previous ones. For example, you apply higher yields, but these are supported by recent observations. In my opinion, due to these two processes, this suggests these previous estimates should be considered underestimates or a lower bound more than your estimate here is an overestimate or an upper bound.

We agree that for these reasons some of the previous estimates could be considered underestimates. For example, Bayer et al 2022 do not consider euphotic zone nitrification. But other previous estimates are likely overstimates: Pachiadaki et al 2017, for example, estimated that NOB alone might fix ~1 PgC/yr. It will be helpful to differentiate these two types of previous estimates. The types that can be connected to our present model (such as Bayer et al 2022, because they use the same yields but not a global model with euphotic zone nitrification, and Zhang et al 2020, because they use the same model but lower yields) should be discussed as such.

We still think that we can consider our model as an upper estimate of global nitrifier C fixation, independent of any relationship with previous estimates or models. We think all of the mechanisms considered result in a maximum potential (or a ~10% overestimation in the case of the export flux) of nitrification and C fixation. However, given this comment and the lack of certainty about euphotic zone nitrification rates, we realize that it is not precise to consider it a true upper bound. We have not suggested, even theoretically, that our estimate should include the maximum amount of euphotic zone nitrification possible. We will change the wording from "upper bound" to an "upper estimate" throughout the revised version.

We have also revised this section as:

**"4.4 An upper estimate of chemoautotrophic C fixation from nitrification**

**The global ecosystem model is a useful tool for estimating global nitrification and associated C fixation rates. The model esti- mate integrates over the wide range in productivity rates across the ocean (Fig. 4), and allows for euphotic zone nitrification to emerge dynamically from microbial interactions. In the range of simulations used in this study, the contribution of nitrification to chemoautotrophic C fixation is 0.2 – 0.5 Pg C $yr^{-1}$. The contribution of AOA (0.2 – 0.3 Pg C $yr^{-1}$) is higher than that of NOB (about 0.1 Pg C $yr^{-1}$).**

**Despite the uncertainties inherent in global ecosystem models, we argue that this estimate constitutes an upper estimate of the chemoautotrophic C fixation rates associated with nitrification. First, the nitrifier C fixation yields input into the model are higher than many previous estimates, and they include the fixed C that is lost to DOC release rather than just that incorporated into biomass (Bayer et al., 2022). Thus, our parameter values provide an upper estimate of the C fixation associated with a given nitrification rate. Second, because the modeled organic export flux is roughly 10% higher than the upper bound of previous estimates, the model may overestimate dark nitrification rates. Third, our simulation includes a significant amount of emergent euphotic zone nitrification, and so 10-30% of nitrifier C fixation is within the euphotic zone in the model. Euphotic zone nitrification is widely observed (Ward, 1987; Dore and Karl, 1996; Ward, 2005;**

**Stephens et al., 2020), though usually not accounted for in biogeochemical models that prescribe light inhibition for nitrification rather than allowing it to emerge from the interactions of dynamic nitrifying populations. For these reasons, the resulting upper bounds of the modeled global C fixation rates (0.34 Pg C yr$^{-1}$ for AOA and 0.12 Pg C yr$^{-1}$ for NOB) are more likely to be overestimates than underestimates."**

Lines 345-347: modeled export flux is larger than previous estimates

It is still unclear to me how this error is accounted for in the uncertainty range. Earlier (e.g. line 281) you show that export production occurs between 11-12 Pg C yr-1 in your model. Is it right that your low-end of your uncertainty range for nitrification rates is driven by a model with export production at 11 Pg C yr-1? Or are the low-end rates reduced in some way to explicitly account for the fact the export production is likely too high? Since this is the clear process why your model estimate is providing an upper bound for global nitrification, I think exactly how you account for likely overestimated export production in your uncertainty range should be explicitly described in the main text. On line 149, you state this will be described in section 3.3.4, but I don't find an explicit description of this other than mentioning that export production is larger than other estimates.

We will explain more clearly explain how the error relates to the export flux. We did not change the parameters that impact the export flux directly. The reported range in export in the global model was due to the choice of cutoff to exclude the very high values of export in the coastal grid points (where the model has no skill and makes wildly wrong predictions; lines 152-153). There is a very clear plateau at this total estimate of export, so it is not an arbitrary cutoff (see below plots). So, all global model simulations have the same 12-13 PgC/yr export estimate. The variation in nitrifier parameter values does not change the export. This is confusing, we realize now! We should more clearly explain this in more places, including the caption of the table.

Second, yes, you are right that the low end of our range of global integrals is indeed with respect to this still very high, constant rate of export. We do not attempt to estimate what nitrification rates would be if the export flux is actually lower, so, our results should not be interpreted as providing the full plausible range of global nitrification rates. This is one reason why we wanted to emphasize that our model provides an upper estimate. Conversely, the low end of our range is not particularly useful. We will clarify this in the revised text. Specifically, we will rewrite the paragraph that includes line 149. We will include a more detailed description of how we deal with the export flux (following our responses here and above), and take out the reference to 3.3.4 (we had meant to refer to the detail on the previous estimates themselves, but we can easily do that in both places).

[Figure]

Fig 1. Histogram of export rates at each horizontal grid box in the model.

[Figure]

Fig. 2. Calculation of global export (cumulative) with the inclusion of higher and higher rates of export. This shows the clear "plateau" at around 12.5 Pg C/yr. The very high but very sparse areas of export are along the coasts, where the model performance breaks down. The blue line gives the calculation if we assume that there is still some export assumed there. Specifically, if we cut off at 10^3 gC/m2/yr, then we assume that all of the locations with higher export than that all have 10^3 gC/m2/yr. The orange line gives the calculation if we simply neglect all of the higher locations.

[Figure]

Fig. 3. A zoom in of Fig. 2 highlighting the range in export (12-13 PgC/yr; blue line) at the "plateau". This is the range that we report, and it is the same in all global model simulations with different nitrifier parameter values.

We have now clarified with the following modified sections of text:

In the Methods section:

**"Additional uncertainty in global nitrification and associated carbon fixation rates exists due to uncertainty in the flux of organic matter exported out of the sunlit surface. As we later clarify in our results, nitrification rates at depth are predominantly set by this export flux, via the supply of $NH^+_4$ from its remineralization. Therefore, we treat the uncertainty due to the export flux by considering that the export flux in our global model is larger (12–13 Pg C yr$^{-1}$) than previous**

estimates (5–11 Pg C yr$^{-1}$; Schlitzer (2000); Henson et al. (2011); Siegel et al. (2014)). In the model, the export flux is controlled by the parameters that dictate the rate of organic matter uptake by heterotrophic microbes combined with the sinking rate of the particulate organic matter (POM). Here, we use the same heterotrophic activity parameterization as in Zakem et al. (2018). Because the model provides an upper bound on the export flux, it provides an upper bound on deep nitrification rates. This is a pragmatic approach because the resulting modeled global rates are lower than many previous estimates. The resulting range of the export flux in Table 3 reflects the choice of cutoff to exclude the very high values of export in the coastal grid points, where the model has no skill. Across the three global model versions, the export flux remained the same with respect to the degree of accuracy represented in Table 3."

In the Results section:

"Due to our heterotrophic activity parameterization, the modeled particulate organic matter export flux (12–13 Pg C yr$^{-1}$, or 2.1–2.4 Pg N yr$^{-1}$, at 1% PAR) is larger than other estimates (5–11 Pg C yr$^{-1}$; Schlitzer (2000); Henson et al. (2011); Siegel et al. (2014)). This choice to overestimate the export flux allows us to provide a meaningful constraint on global rates. Because deep nitrification rates are set by the export flux of organic nitrogen, the upper bound on the export flux provides an upper bound on dark nitrification rates. Variation in nitrifier parameter values does not significantly change the export flux. In all models, the export flux remained the same with respect to the degree of accuracy represented in Table 3."

Lines 353-358: comparison with Baltar and Herndl (2019) estimate

It seems to me that comparing your nitrification only estimate with a total deep ocean carbon fixation is a little like comparing "apples to oranges". I'm not familiar with that Baltar and Herndl study, which apparently provides a very large range, so I'm wondering if it is possible to infer a first-order estimate of the nitrification contribution from that study. If you believe that nitrification only accounts for a small fraction of total deep carbon fixation, is there any other specific metabolism you think may be most important to explore next?

Yes, we agree that it is an "apples to oranges" comparison because potentially other metabolisms are responsible for deep C fixation. In the revised version, we'll put the second sentence of this paragraph first, so that at first read it doesn't seem as if we are suggesting that the Baltar and Herndl paper is wrong, but rather, that our studies combined imply that there are unaccounted-for metabolisms. It is a great suggestion to include some specific possibilities, such as sulfate reduction, and we will add this to the revised version.

This paragraph has been modified as:

"It is possible that metabolisms other than nitrification contribute to deep ocean C fixation. Our upper estimate of the modeled C fixation rate from nitrification (0.5 Pg C yr$^{-1}$) is substantially less than the 1–10 Pg C yr$^{-1}$ of deep carbon fixation estimated by Baltar and Herndl (2019). Given that nitrification rates wane sharply with depth in the dark ocean (Ward (1987); Dore and Karl (1996); Newell et al. (2013); Zhang et al. (2020); Figs. 2 and 3), the contribution of nitrification to deep C fixation may decrease substantially with depth (Pachiadaki et al., 2017). As nitrification wanes, a diverse microbial community carrying out other metabolisms, such as sulfate reduction, may dominate C fixation rates (Swan, 2011)."